# Development of a sequencing system for spatial decoding of DNA barcode molecules at single-molecule resolution

Yusuke Oguchi [1,2,3✉], Hirofumi Shintaku [2] & Sotaro Uemura[1✉]

Single-cell transcriptome analysis has been revolutionized by DNA barcodes that index cDNA libraries, allowing highly multiplexed analyses to be performed. Furthermore, DNA barcodes are being leveraged for spatial transcriptomes. Although spatial resolution relies on methods used to decode DNA barcodes, achieving single-molecule decoding remains a challenge. Here, we developed an in-house sequencing system inspired by a single-molecule sequencing system, HeliScope, to spatially decode DNA barcode molecules at single-molecule resolution. We benchmarked our system with 30 types of DNA barcode molecules and obtained an average read length of ~20 nt with an error rate of less than 5% per nucleotide, which was sufficient to spatially identify them. Additionally, we spatially identified DNA barcode molecules bound to antibodies at single-molecule resolution. Leveraging this, we devised a method, termed "molecular foot printing", showing potential for applying our system not only to spatial transcriptomics, but also to spatial proteomics.

[1] Department of Biological Sciences, Graduate School of Science, The University of Tokyo, 7-3-1 Hongo, Bunkyo, Tokyo 113-0033, Japan. [2] RIKEN Cluster for Pioneering Research, 2-1 Hirosawa, Wako, Saitama 351-0198, Japan. [3] JST, PRESTO, 4-1-8 Honcho, Kawaguchi, Saitama 332-0012, Japan. ✉email: yusuke.oguchi@riken.jp; uemura@bs.s.u-tokyo.ac.jp

Sequencing technologies have undergone massive transformations during this decade[1], allowing researchers to perform single-cell transcriptome analyses relatively easily, even for tens of thousands of cells at a time[2–5]. Besides sequencing instruments, molecular barcodes, which offer highly quantitative gene counting while eliminating amplification biases[6], act as a key to high-throughput and multiplexed analyses[2–5]. Furthermore, in addition to spatial transcriptomics[7,8] molecular barcodes have contributed to multiomics analyses[9] and visualizing of surface proteins[10]. However, existing spatial transcriptome analyses involve a trade-off between spatial resolution of gene position (single-molecule resolution or not) and the number of target genes (genome-wide or not). For example, slide-seq[8] achieved genome-wide analyses via DNA barcoded beads; however, its spatial resolution is limited to that of the size of a cell, due to being governed by the size of DNA barcoded beads (~10 μm). Hence, decoding single-molecule barcodes remains a challenge. The current study developed a sequencing system to spatially decode DNA barcode molecules at single-molecule resolution, by repurposing the single-molecule sequencing system, HeliScope[11].

For most researchers, application of sequencers is limited to objectives that were originally defined by the manufacturers, that is sequencing DNA/cDNA molecule libraries. However, several groups have repurposed existing sequencers for experiments of their own design[12]. For instance, an old Illumina sequencer (GAIIx) was repurposed to study protein and nucleic-acid biochemistry on a massive scale[13–15]. Additionally, a PacBio sequencer was repurposed to visualize protein translation of the ribosome at physiologically relevant micromolar ligand concentrations[16]. Although these have been achieved by researchers through tremendous efforts[12], such repurposing has remained at the imaging device level, leaving a large gap in the ability of individual researchers to apply them under industry-free control, since detailed information, including sequencing reaction/reagents, are not being disclosed. To overcome such obstacles, we repurposed HeliScope to achieve our intended experimental goals, without dependence on the original manufacturer for components of the associated enzymatic reactions.

HeliScope, which was originally developed and commercialized by Helicos BioSciences, offers unbiased DNA sequencing[11,17] and direct RNA sequencing[18], and was leveraged for large-scale research[19] by the international consortium, FANTOM. It demonstrated potential for low-quantity/attomole-level DNA/cDNA sequencing applications[20], such as ChIP-seq[21] and circulating cell-free blood nucleic acids[22]. These results were attributed to a single-molecule sequencing-by-synthesis (smSBS) that HeliScope uniquely achieved with Virtual Terminator (VT) nucleotides[23]. VTs are nucleotide analogs containing a chemically cleavable group that prevents the addition of another nucleotide and carries a fluorescent dye, allowing smSBS to obtain sequence information of individual DNA molecules and their locations. Therefore, this method appeared to be the best out of currently available methods, for achieving spatial decoding of DNA barcode molecules at single-molecule resolutions. However, it was found that HeliScope was not specialized for this purpose. Furthermore, it was not amenable to modifications that were required for our experimental purposes.

Therefore, the current study constructed and benchmarked our own smSBS, and identified DNA barcode molecules at a single-molecule resolution. For further benchmarking, we analyzed a minute amount of cDNA library, comparable to that obtained from a single cell, and analyzed it using an Illumina sequencer, showing the unique potential of our system for gene quantification. Additionally, as an improvement from the original, a sample capture method was developed using biotin-avidin interactions

that was comparable to that achieved with a covalent binding-based method, which enables samples to be captured via "hybridization-free sample immobilization". Furthermore, we identified DNA barcode molecules bound to antibodies used in CITE-seq[9] and analogous molecules used in CODEX[10]; unlike in previous studies[9,10], these were identified with spatial information at a single-molecule resolution. Finally, we demonstrated a method for spatial analysis, termed "molecular foot printing", that allows DNA barcode molecules labeled on the surface of cells and/or within cells to be transferred onto a sequencing flow cell, and subsequently sequenced.

## Results

### Development of in-house single-molecule sequencing system.
With reference to previously described methods[24,25], we reconstructed the sequencing system (Fig. 1a) that performs smSBS (Fig. 1b), by combining a commercially available TIRF microscope and fluidic control pumps (Methods section). We also constructed a unique flow cell with a 1-μl volume applicable to a few microliters of samples, which is easily modifiable. For instance, sample capture oligos are readily replaced to hybridize a specific sample.

To validate VT incorporation, we captured a dense 1 nM sample from an oligonucleotide molecule on a substrate (Supplementary Fig. 1a, b), which appeared as a bright single fluorescence image, allowing us to easily determine whether the incorporations were correct (Supplementary Fig. 1b). Subsequently, we succeeded in determining the sequence of individual molecules at a lower capture density achieved with 25 pM (Supplementary Fig. 1c; single-molecule detection, and stage drift compensation; Methods section). We were able to prepare all required components for smSBS, including not only instruments but also sequencing reagents, independently of Helicos BioSciences.

### Barcode decoding at single-molecule resolution.
Our system identified 30 types of DNA barcode molecules captured on a flow cell of 1 μl at 25 pM in total (Fig. 1c and Supplementary Table 1). First, we visualized all molecules simultaneously, at a position termed the 1st position, by adding virtual terminators (Fig. 1d and Supplementary Fig. 2a). We then performed a sequencing cycle consisting of 24 quads, wherein one round, which incorporates four respective nucleotides, is termed a "quad (Q)". Although the total area of sample capturing and scanning of the flow cell was 18.5 mm × 1.5 mm, corresponding to ~4000 fields of view (FOV), of size 75 μm × 75 μm, only 16 FOVs were scanned for barcode identification, which took ~2 h per quad and ~2 days in total. This run, yielded sequence reads of lengths that were sufficient to enable identification of all 30 types of molecules (with BLAST, e-value <0.01; Fig. 1d, e), indicating high reproducibility of results related to separate identifying FOVs (Supplementary Fig. 2b).

Next, we traced VT incorporations of each individual molecule (Fig. 1f). Although most molecules tolerated elongation up to 24Q (the last cycle; Supplementary Fig. 2c), some stopped elongation before this point (Supplementary Fig. 2d, e). Compared to identified reads, unidentified reads tended to terminate elongation at relatively earlier sequence cycles, most frequently at ~6Q (Supplementary Fig. 2f), which is shown by the differences between averaged elongation traces (Fig. 1f). Therefore, unidentified reads are primarily attributed to short reads that stopped at earlier sequencing cycles.

The average read length increased with sequencing cycles (Fig. 1g), however, the rate of increase slowed, particularly after 18Q (8.1, 13.7, 18.1, and 21.2 nt on average for 6, 12, 18, and 24Q, respectively). Efficiency of barcode molecule identification

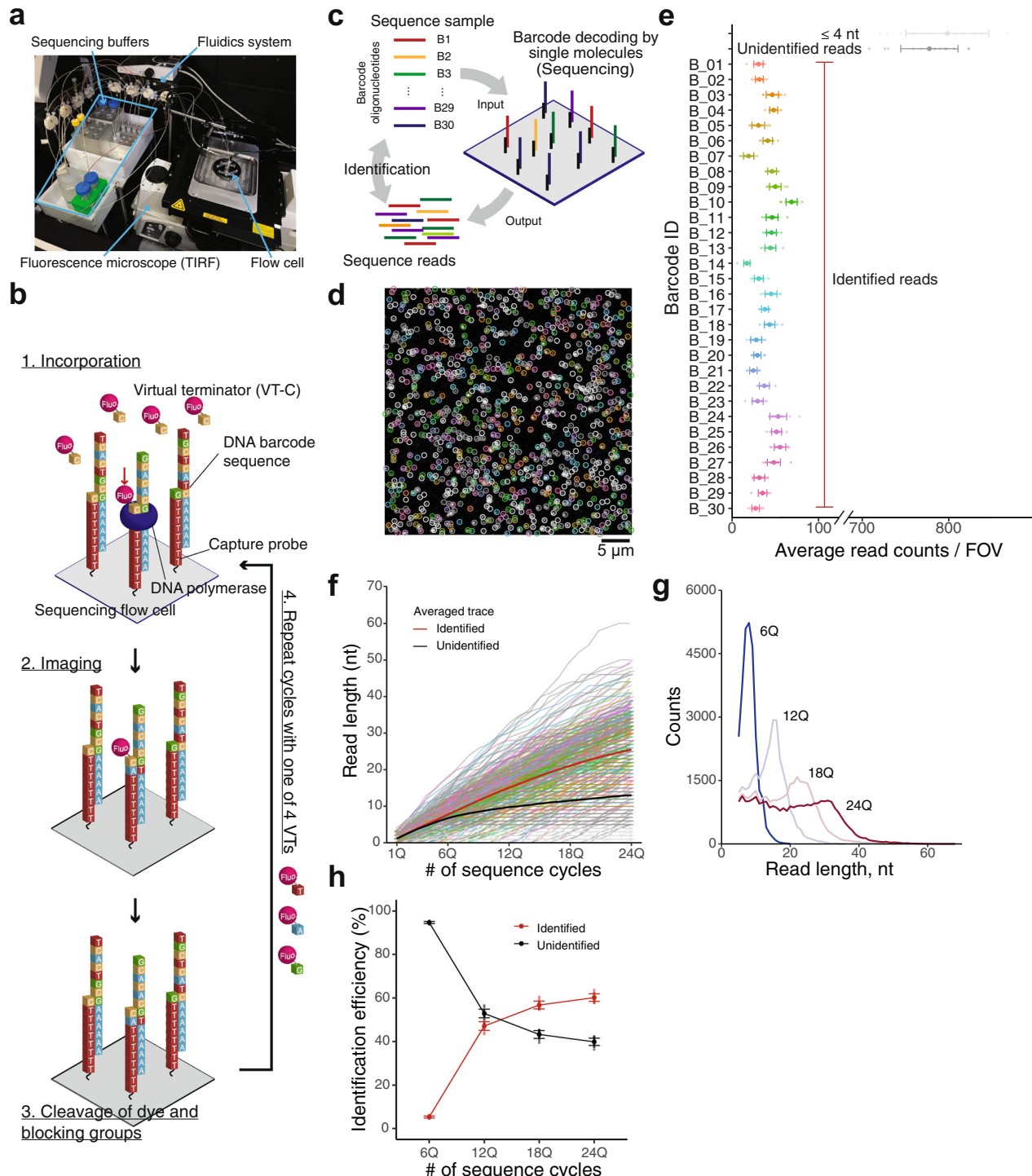

increased with the number of sequence cycles (Fig. 1h). The results also indicated that a read length obtained by 18Q (18.1 ± 7.9 nt) is sufficient for identification of ~30 types of barcode molecules. Furthermore, we used alignment results to calculate incorporation accuracy per base, which estimated accuracy to be approximately 95% (Supplementary Fig. 3a), comparable to that obtained from the original HeliScope[20]. We also analyzed the accuracy of 2nd incorporation in two subsequent incorporations (Supplementary Fig. 3b–d), which demonstrated slight dependence on the 1st base, while the error rates of the same two base incorporations (AA, CC, GG, and TT) were higher than those of the other combinations.

**Analyzing a minute amount of cDNA library derived from K562 cells, using our system**. Our findings indicated the potential of our smSBS to analyze a small quantity of cDNA (~10 pg) prepared from cells (Supplementary Fig. 4a). A few tens of pg of full-length cDNA molecules[26] synthesized from ~10 K562 cells, using a SMART-seq v4 kit, were evaluated. Notably, fragmentation was required before these molecules could be measured with an Illumina sequencer. However, we were able to detect cDNA molecules that retained their full-length. We bound capture oligos with the sequence of a template switching oligo (TSO) to the surface of our flow cell. As both 3' ends of the 1st and 2nd cDNA molecules had a complementary sequence of

**Fig. 1 Spatial decoding of DNA barcode molecules with the in-house single-molecule sequencing system. a** A photo of our single-molecule sequencing system. **b** Schematic illustration of single-molecule sequencing-by-synthesis utilized in our system. In each cycle, one of four virtual terminator nucleotides (VT-C, VT-T, VT-A, and VT-G) is incorporated (1), and unincorporated VTs are washed, followed by fluorescence imaging (2). The fluorescent dye and inhibitory groups are then removed using TCEP to permit the addition of the next virtual terminator (3). This cycle is repeated in the VT-C, VT-T, VT-A, and VT-G order (4). **c** Schematic of DNA barcode identification test; 30 types of DNA barcode molecules were tested. A poly-A tail was added to these barcodes to allow them to be captured on the flow cell to which $dT_{50}$ capture oligos were attached. Obtained sequence reads were identified with BLAST. **d** Fluorescence image of individual barcode molecules visualized by the 1st virtual terminator incorporation. Identified molecules are encircled with colors corresponding to barcode IDs as shown in **e**. Scale bar represents 5 μm. **e** The average number of identified barcode molecules per field of view (FOV). Error bars represent SD ($N = 16$, 16 of FOVs were observed for this test). Unidentified reads indicate that those reads were not identified in the reference or identified while e-values of BLAST were not <0.01. **f** Extension trace of individual molecules. Each trace represents the extension of each molecule shown in **d** with increasing numbers of sequencing cycles. Ideally, at least 1 nt should be extended by each cycle. Colors of individual traces correspond to those shown in **e**. Average trace for identified (red) and unidentified (black) was estimated as the average of the traces categorized as identified and unidentified, respectively. **g** Comparison of read length distribution between the numbers of sequencing cycles; reads <5 nt were excluded. **h** Identification efficiency against the number of sequencing cycle. Identification efficiency was calculated by individual FOVs ($N = 16$); error bars represent SD; reads <5 nt were excluded from this analysis.

TSO, which was added during cDNA synthesis, they were captured on a flow cell via TSO capture oligos (Fig. 2a).

Of the 132 ng of amplified cDNA from ~7.5 cells, 43 pg (1/3000 of total amount) was loaded onto a sequence flow cell (Supplementary Fig. 4a). Then, 24Q sequencing was conducted with a scanning area equivalent to 1/300 of whole FOVs, achieved a non-biased capture density irrespective of FOVs (Supplementary Fig. 4b) and obtained 397,535 reads (>4 nt). However, ~30% of these were byproducts (read containing TSOs, complementary of TSOs, etc, Supplementary Table 2) presumably produced during cDNA synthesis and/or following PCR amplification. Byproducts were identified by mapping all reads to a predicted byproduct reference, resulting in 129,487 predicted byproduct reads. Meanwhile, of the 186,626 reads that were longer than 18 nt and that did not contain byproducts, 173,051 mapped to the human genome with a 92.7% mapping rate (29.2% unique) using STAR. We attributed unmapped reads primarily to long reads (>60 nt; Fig. 2b). The number of genes counted via HTseq (count > 0) from our system was 7148, of which 4366 were co-detected via Illumina (RNA-Seq via Expectation-Maximization (RSEM), TPM > 0; Fig. 2c), showing a correlation coefficient of 0.53 ($\log_{10}$ (count + 1) vs. $\log_{10}$ (TPM + 1)); (Fig. 2d). Given the input amount (1/3000 of total) and the scanned area (1/300 of total), we detected 7148 genes from an amount of cDNA equivalent to a ~$1/10^6$ of the total amount synthesized and amplified from 7.5 cells (Supplementary Fig. 4a). Moreover, it was noted that Illumina data, compared with that of smSBS, were down-sampled into 200k paired-end reads (Supplementary Fig. 4c; down-sampling analysis).

Since full-length cDNA molecules are directly captured on a substrate, capture efficiency may differ depending on length. However, we found that the length of detected genes spanned from 59 nt to 39,314 nt, which was comparable to those obtained with an Illumina sequencer (109 nt–22,743 nt) detecting fragmented cDNAs (Fig. 2e). In addition, we obtained unique gene coverage of biased 5′ and 3′ ends derived from 1st and 2nd strands, respectively (Fig. 2f), which are generally characterized as low coverage using fragmented cDNA analyzed with Illumina-seq. Furthermore, we used $dT_{25}VN$ as a capture oligo targeting only the 2nd strand, resulting in a coverage of only 3′ ends as expected (Supplementary Fig. 4d). Here we also measured an amplification-free sample from a single cell (Supplementary Fig. 5). However, we were unable to detect a significant number of genes. Thus, our system is applicable to ~10 pg of cDNA input and offers unique gene detection of full-length cDNA molecules without fragmentation by detecting either the 5′ or 3′ end region, which may contribute to eliminating sequencing biases produced by the fragmentation process[27–30].

**The effect of binding stability among capture oligos and the flow cell surfaces on read length.** Next, we examined binding stability between capture oligo and the flow cell surface, as sequence cycles must be repeated to obtain longer reads, while samples should be anchored on the flow cell ideally over the entire sequencing period. We compared capture with covalent bonds via NHS and $NH_2$-group reactions and biotin-avidin interactions (Fig. 3a). Regarding the biotin-avidin bond, we observed the effect of the number of biotins molecules attached to a capture oligo on binding stability (1 or 4 biotins per capture oligo). In regard to the sample tested, the same 30 oligo types mentioned above were used (Fig. 1).

Each incorporation trace of individual molecules was extracted and averaged (Fig. 3b–d), NHS exhibited the highest elongation efficiency and biotin × 1 the lowest. Also, the average read length following the 24Q sequencing run is shown (Fig. 3e). As indicated by elongation efficiency, the NHS bond average read length was the longest followed by biotin × 4 and biotin × 1 (22.2 ± 11.1, 18.6 ± 9.9, and 14.1 ± 7.7 nt (mean ± s.d, combined among technical replicates), respectively. Differences between all groups were statistically significant ($p < 2.2 \times 10^{-16}$; $U$-test). We further investigated the number of quads at which extension stopped (Fig. 3f), showing that biotin × 1 stopped incorporation at an earlier quad. The temporal pausing frequency showed no difference between all three conditions (Supplementary Fig. 6), indicating that lower elongation efficiency may be attributed to full stop events (Fig. 3f and Supplementary Fig. 2d, e), which may correspond to the detaching of molecules from the surface.

Barcode quantification showed high reproducibility between technical replicates (Fig. 3g; $r > 0.85$) and relatively high correlation between different conditions (Fig. 3g; $r > 0.75$). Thus, we showed sample capturing and sequencing with non-covalent bonds, which were able to extend their stability on the surface to levels comparable with that via covalent bonds, by increasing the number of biotin-avidin interactions per capture oligo.

**An alternative sequencing polymerase.** Additionally, since sequencing polymerases are key players in performing smSBS with high accuracy, we also sought to identify a polymerase capable of correctly incorporating VTs with higher efficiency than the original polymerase. In fact, Therminator DNA polymerase™ (ThI), an applicable polymerase, was applied in all experiments presented herein. We also performed smSBS with Klenow fragments (exo-, KF) used in a previous study[24], to compare two polymerases. Although both polymerases detected all 30 types of DNA barcode sequences, the detection sensitivity of ThI was slightly higher than that of KF, as is evident in the detection

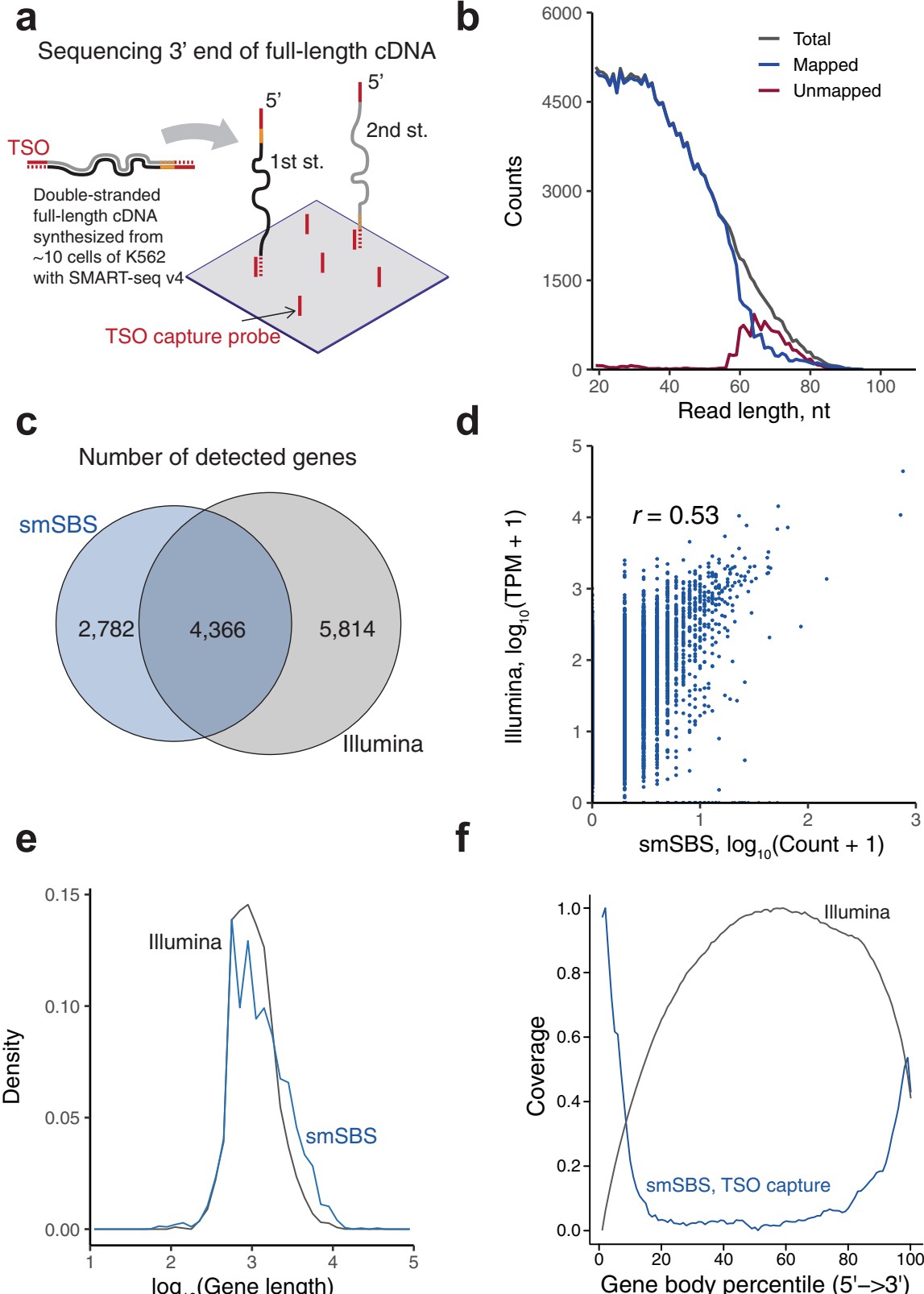

**Fig. 2 Application of smSBS to a small quantity (~10 pg) of cDNA library of K562 cells. a** Schematic illustration for the capture and sequence of unfragmented full-length cDNAs. Lines in the cDNA molecules indicate TSO (red solid), complementary of TSO (red dashed), polydT (yellow solid), and poly-A (yellow dashed). **b** Read length distribution after 24Q sequencing; reads <19 nt and containing byproducts were filtered out. **c** Venn diagram of the number of genes detected with smSBS and an Illumina sequencer (Hiseq2500). Genes detected with smSBS and Illumina were counted as HTseq-count > 0 and TPM of RSEM > 0, respectively. **d** Comparison of gene expressions obtained via smSBS and Illumina. **e** Normalized density plot of gene length detected via smSBS and Illumina. **f** Gene body coverage via smSBS and Illumina. Note that, the Illumina results shown in **c**–**f** were calculated with the reads down-sampled into 200k paired-end reads.

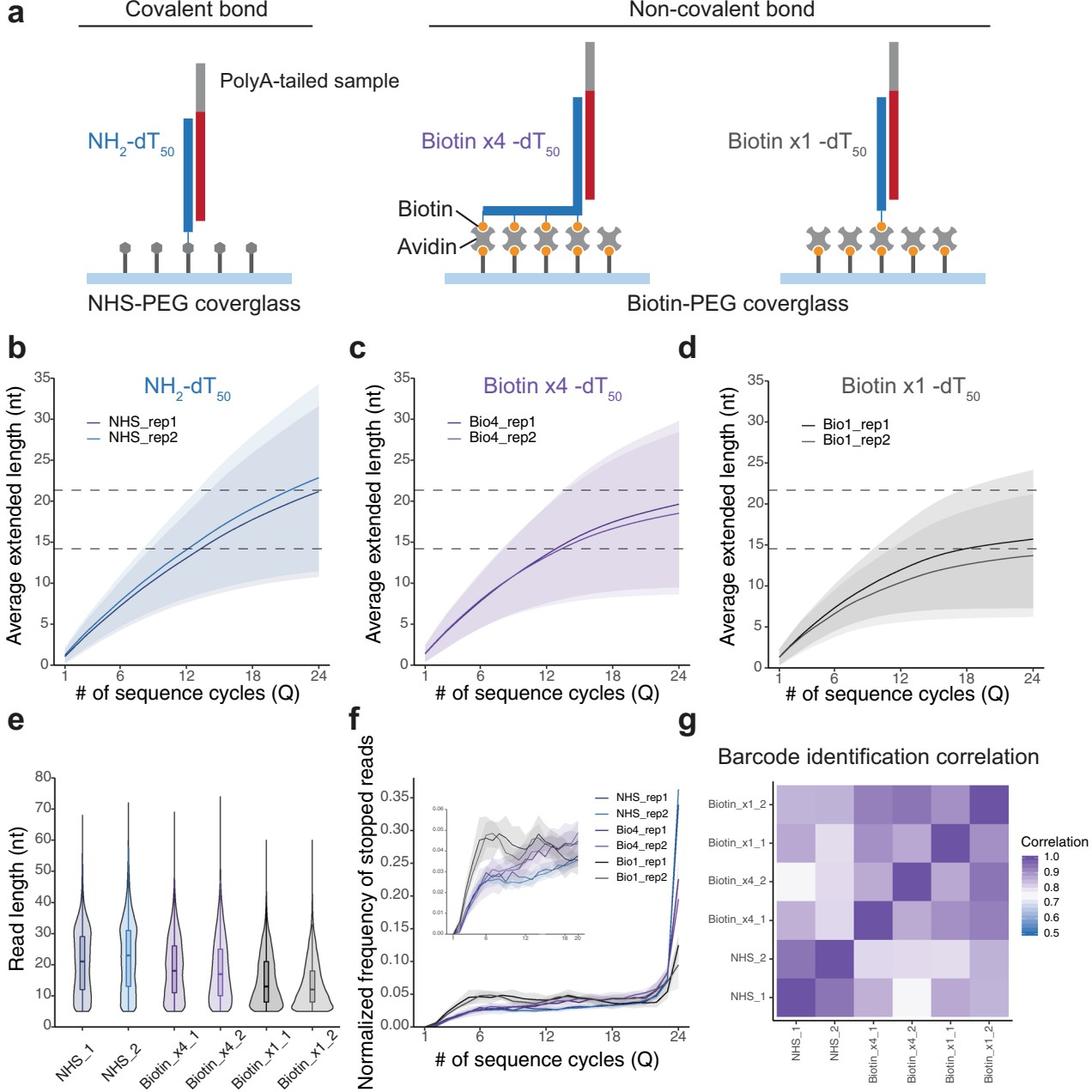

**Fig. 3 Impact of anchoring methods used to capture oligos to the flow cell on sequence performance. a** Schematic illustration of flow cells. Three different types of binding between capture probes and a flow cell surface were tested as follows; a covalent bond achieved via click chemistry NHS and $NH_2$ reaction, a non-covalent bond achieved via biotin-avidin but different in the number of biotins per probe, four and one biotin molecules per probe, are shown in the left to the right panels respectively. The same barcode sample shown in Fig. 1 was used for this test. **b–d** Averaged extension traces for NHS (**b**), biotin × 4 (**c**), and biotin × 1 (**d**), calculated in the same manner shown in Fig. 1f. Colored ribbon plots represent SD at each cycle. All three conditions were repeated twice. **e** Read length distributions; reads <5 nt were filtered out. **f** Normalized frequency of stopped reads at each cycle. Note that the value at 24Q corresponds to the fraction of reads that reached 24Q (we were unable to identify whether these reads stopped or extended after this point as 24Q was the last cycle). Colored ribbon plots represent SD. **g** Cross-correlation of the detected count of 30 types of barcodes between conditions.

difference of B_09 (Supplementary Fig. 7a). Furthermore, reproducibility was high between technical replicates, however, slightly lower between different polymerases (Supplementary Fig. 7b), implying the possibility of biases in detection attributed to the polymerase. However, the degree of incorporation accuracy was ~95% (correct incorporation in Supplementary Fig. 8) with a slightly different breakdown of errors (Supplementary Fig. 8). Meanwhile, the average read length was slightly longer with ThI than with KF (Supplementary Fig. 7c), which was attributed to the difference in the pause rate (frequency of blank quads,

Supplementary Fig. 7d), rather than the full stop rate (Supplementary Fig. 7e). Thus, we successfully identified an alternative polymerase for smSBS, and although there were slight differences in the polymerases, ThI more readily incorporated virtual terminators (having a higher incorporating rate) compared to KF, while the incorporation accuracy between the two was similar.

**Identification of DNA barcoded antibodies at single-molecule resolution.** Here, we decoded DNA barcoded (15 nt in length) antibodies using CITE-seq, which are commercially available

(TotalSeq™, biolegends), at single-molecule resolution (Fig. 4a). Although we were able to capture and sequence the antibodies in a more straight forward manner via adding a poly-A tail to the barcode oligonucleotide (Figs. 1 and 3), we also sought to demonstrate proof of concept for our "hybridization-free sample immobilization" following sequencing (Fig. 4a and Supplementary Fig. 9). To demonstrate this, we selected an anti-CD55 antibody tagged with CD55 barcode molecules (TotalSeq™-A0383 anti-human CD55 Antibody) and determined whether they were correctly identified.

An image of the 1st position (position of molecules) was obtained (Fig. 4b). Although sequencing occurred in a downward direction (Fig. 4a, rightmost), we observed successful elongation (Fig. 4c) and obtained reads having average read lengths of $15.4 \pm 5.9$ nt ($n = 33{,}497$); (Fig. 4e), which were comparable to those with upward sequencing. Approximately 60% of all reads, with the exception of those <5 nt, were identified as CD55 with BLAST e-values < 0.1 (Fig. 4d, f), where reads were identified by mapping them to a reference containing only a CD55 sequence. Furthermore, molecules that showed no incorporation events (0 nt in

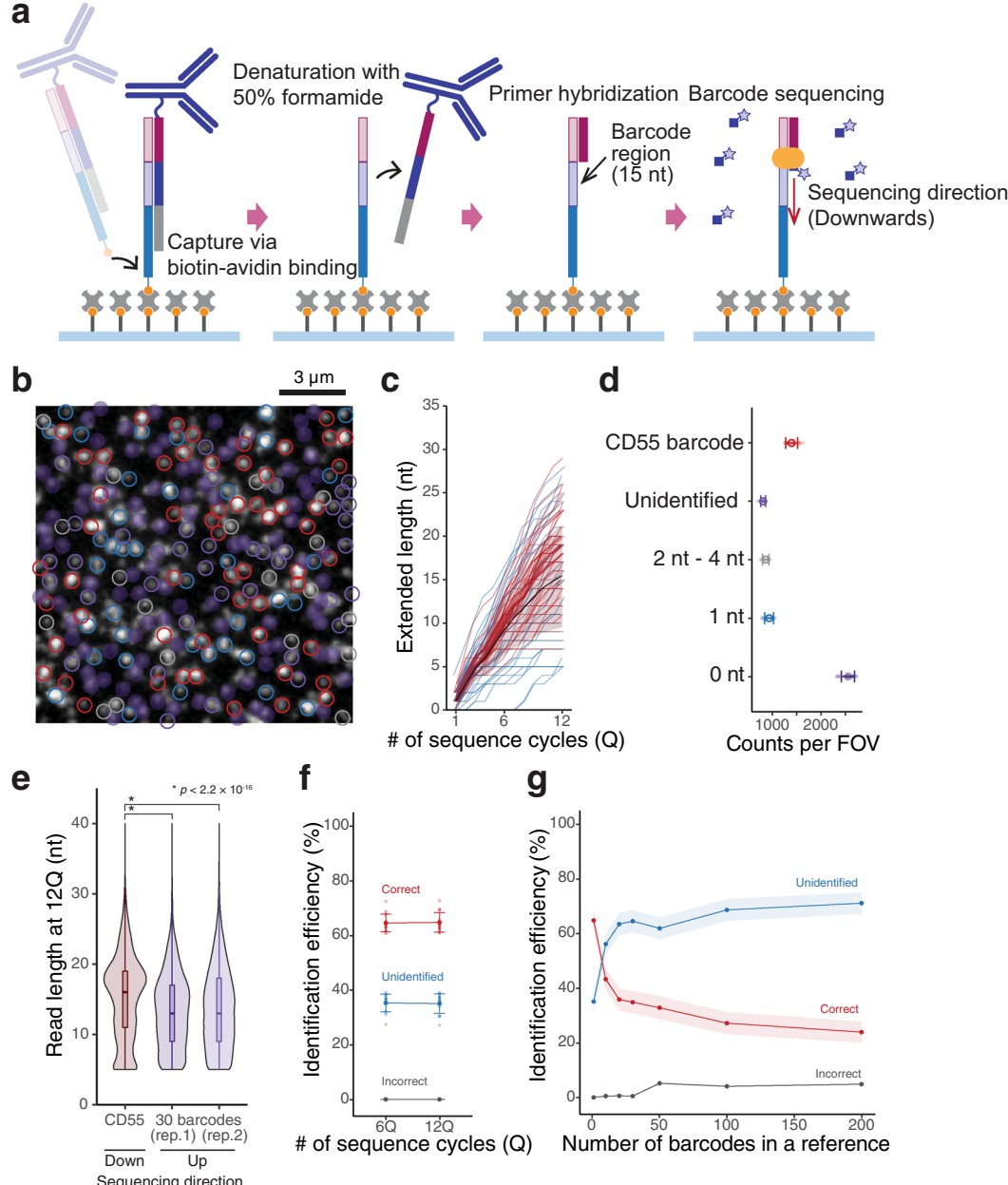

**Fig. 4 Visualization and identification of DNA barcoded antibodies at single-molecule resolution. a** Schematic illustrations showing a procedure of sequencing of DNA barcoded antibody (Ab-Oligo) on our system. **b** A fluorescence image of individual CD55 Ab-Oligos visualized by the 1st virtual terminator incorporation. Scale bar represents 3 µm. Molecules identified as single fluorescence spots via image analysis are encircled with colors corresponding to those shown in **d**. **c** Extension traces of individual molecules; reads <5 nt were excluded. Each trace represents the extension of each molecule shown in **b**. **d** Average number of identified barcode molecules per FOV. Error bars represent SD ($N = 16$). Note that for analyses shown in **b**–**d**, **f** reference to identify individual reads contained only CD55 barcode sequences. **e** Comparison of read lengths between different sequencing directions. **f** Comparison of identification efficiency between 6Q and 12Q. **g** Estimation of multiplexing ability. False positive identification (incorrect) was calculated by changing the number of barcode sequences in the reference (1–200 types of sequences). Up to 30 types in a reference, false positive identification was negligible. Colored ribbon plots represent SD.

Fig. 4b, d), and instead showed non-specific binding during the 1st VT incorporation, were frequently observed (Fig. 4d; 0 nt), which implied reduced blocking of non-specific binding of the substrate during the procedure (Fig. 4a).

Further, we estimated how many targets could be effectively multiplexed by changing the number of barcodes in the reference (Fig. 4g), and found that up to 30 targets could be simultaneously processed with fewer false positives (incorrect identification, Fig. 4g, gray line). For further multiplexing (>30 targets), further improvements, such as increasing barcode lengths beyond 15 nt, may be required. However, we succeeded in spatially decoding DNA barcoded antibodies at single-molecule resolution.

### A method toward spatial analysis, "molecular foot printing".
Finally, by leveraging hybridization-free sample immobilization, we demonstrated a method toward spatial analysis, termed "molecular foot printing" (Supplementary Fig. 9). This strategy allows DNA barcode molecules labeled on the surface of cells and/or within cells to be transferred onto a sequencing flow cell, and subsequently sequenced (identification and visualization of molecules). As a proof of concept experiment, we first demonstrated this with protein-G coated beads mimicking cells (Fig. 5). Beads binding a DNA barcode antibody complex (TotalSeq™-A0361 anti-human CD59 antibody hybridized to the complementary sequence of the barcode sequence, Fig. 5a) were introduced into a sequencing flow cell. We confirmed that the complementary strands of the barcode molecule were successfully transferred onto the flow cell surface from the beads, while the beads were washed off the sequence flow cell. Further, we conducted sequence cycles and observed a correct order of VT incorporation according to the barcode sequence. Thus, we were able to transform DNA molecules binding with an object to a sequencing flow cell and showed that these transformed DNA molecules were effectively sequenced with our system. Furthermore, we performed a similar experiment with K562 cells (Supplementary Fig. 10), confirming that the complementary probes of the anti-CD55 barcode molecules on a K562 cell were transferred onto a sequencing flow cell surface.

### Discussion
Fluorescence microscopy is a powerful tool for biological research, however, the ability to observe multiple objects simultaneously (multiplex) is limited by the number of spectrally distinguishable fluorophores. To overcome this limitation, several approaches have been devised by leveraging DNA barcoding technologies[31–37], some of which offer simultaneous labeling of target molecules with orthogonal DNA barcoded affinity reagents[32], followed by sequential imaging via hybridization of dye-labeled complementary oligos[33,34]. As an alternate approach temporal barcodes have been designed that do not rely on spectral information of the dye molecules but rather exploit distinct temporal fluorescence intensity signals produced via hybridization kinetics of dye-labeled complementary oligos[35–37]. Although this approach has significantly improved the multiplexing ability compared to conventional fluorescence microscopy, target specific probes are still required, which will ultimately limit the multiplex capacity of the system.

Furthermore, DNA barcoding technologies have also recently been applied for spatial transcriptome and proteome analyses. For this purpose, it is useful to decode barcode molecules using a sequencing-by-synthesis approach. For instance, although CODEX[10] is a fluorescence imaging-based technique, using DNA barcode molecule tagged antibodies, in place of conventional fluorophore tagged antibodies, and decoding them via a sequencing-by-synthesis offers highly multiplexed surface markers with which cells may be identified. Presently, these spatial resolutions correspond to specific cell sizes. However, our smSBS would allow visualization of individual molecules and increase the number of targets to be multiplexed with a high degree of accuracy in identification, thereby advancing such analyses. Although in the current study we demonstrated that our system effectively multiplexes up to 30 targets, we were able to further estimate the multiplexing capacity using a simple model (Supplementary Fig. 11a–f). This indicated that our system could be potentially applicable for several hundred or thousand unique molecules. An example (with reasonable errors) for 10,000 unique molecules matching our empirical data distribution is shown in Supplementary Fig. 11g and h. Although seqFISH+[38], a spatial transcriptome technology, has recently succeeded in visualizing gene positions at single-molecule resolutions of up to 10,000 genes simultaneously in single cells, the number of target genes remains limited. In contrast to target specific technologies, Slide-seq[8] and HDST[39] conducted genome-wide expression analyses using DNA barcoded beads. However, spatial resolution was limited to bead sizes of 10 μm and 2 μm, respectively. Our smSBS improved spatial resolution by decoding DNA barcodes at single-molecule resolution. Comparable to our method, Barista-seq[40] and INSTA-seq[41] also decode DNA barcode molecules by individual molecules leveraging in-situ sequencing, however, require amplification (rolling amplification) of barcode molecules, which leads to inefficient decoding and spatial resolution[40].

Further, our platform has the capacity to exploit conventional transcriptome analysis for detection of minute amounts of samples. Although sequencers utilizing SBS have been targeted for high-throughput and cost-effective analyses, some Illumina sequencers have been designed for small-scale analyses, such as personal benchtop scale analyses including MiSeq, MiniSeq, and iSeq. However, these analyses require a relatively large amount of samples (at least 300 pg for MiniSeq for estimating with 500 μl of 1.8 pM ds-cDNA having 300 bp on average). Although a nanopore aided ZMW sequencing system[42] improved capture efficiency compared to the original system, the amount of applicable sample (a few hundred pg) was still limited. Compared to the above systems, ours is applicable to ~10 pg of full-length cDNA input detected at a density of ~3000 molecules/FOV (Supplementary Fig. 4b). Further, this could be reduced by at least one-tenth as we obtained sequencing reads with a sample of a lower concentration, achieving a density of ~100 molecules/FOV (Supplementary Fig. 5b). This system is expected to ultimately achieve amplification-free sequencing even with cDNA molecules synthesized from a single cell. We attempted to detect cDNA molecules immediately following cDNA synthesis from a single cell with SMART-seq v4, however, the system was only able to detect TSO sequence-like byproducts. This was attributed to a low concentration of cDNA molecules of ~0.1 pg (estimated as 1% of mRNA in a 10 pg of total RNA extracted from a single cell). To detect genes from a 0.1-pg sample, an area 100 times larger than that for 10 pg may have to be scanned, since smaller amounts yield lower capture densities. The rate-limiting step of our system is considered to be imaging, and hence is not practically applicable. We envision that further improvements, such as increased sample concentrations and fragmentation to increase molecular density, are required.

In summary, the system described herein not only reconstructed smSBS by applying sequencing chemistries inspired by the original Helicos BioScience system, but also introduced further improvements such as barcode decoding, small quantity and full-length cDNA capturing and sequencing, as well as biotin-based durable sample capturing. In regard to novelty of application, we succeeded in identifying DNA barcode molecules at single-molecule resolution with our in-house barcode molecules, as well as commercially available antibody-tagged barcode molecules. We also provided proof of concept experiments for

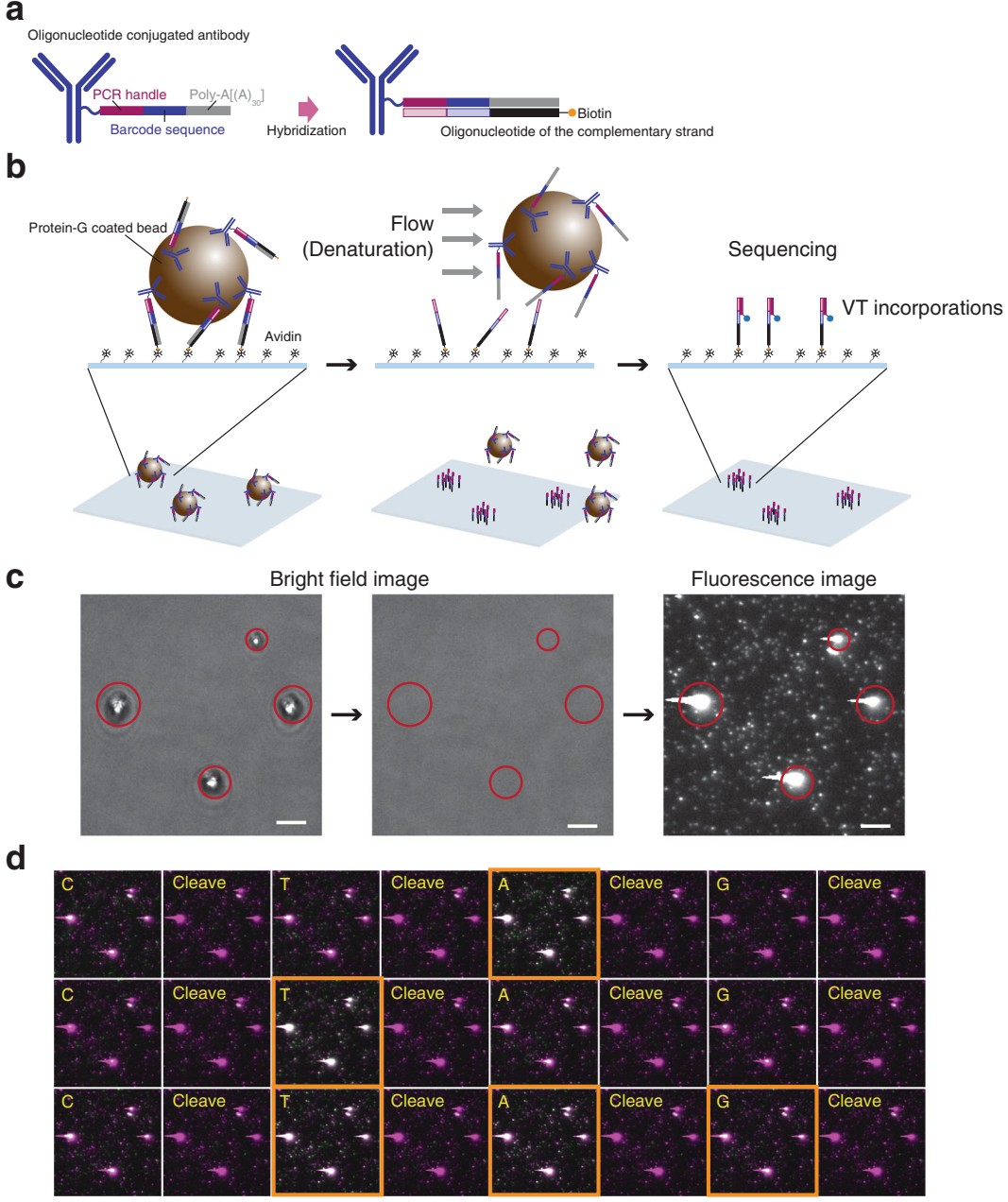

**Fig. 5 A proof of concept experiment for molecular foot printing. a** Ab-barcode complex with a pre-synthesized oligonucleotide of the complementary strand. Barcode-strand was double-stranded with an oligonucleotide of the complementary sequence of the barcode instead of reverse transcription. **b** Schematic illustration of the experiment. Protein-G coated magnetic beads binding with the double-stranded Ab-barcode were introduced into a sequencing flow cell and captured on the surface via biotin-avidin interactions (left). After observing the beads, they were washed from the flow cell via a pressure driven flow of 50% formamide solution, while the complementary sequences remained on the surface (middle). Visualization of the complementary sequences remained on the surface by incorporating VTs (right). **c** Microscopic images corresponding to the schematic illustration shown in **b**. Blight filed images of protein-G coated beads on the surface (encircled in red, left panel). After washing, the beads were confirmed to have moved from the original position (middle). Fluorescence image of the complementary sequences remained on the surface visualized by VT incorporations (right panel). Scale bars shown in **c** indicate 5 μm. **d** Overlaid images of each cycle (green) on the image of the 1st cycle (magenta). Rectangles in orange indicate expected cycles of VT incorporation according to the barcode sequence. Sequencing proceeded from left to right (CTAG) and top to bottom (1Q to 3Q) in this sequential image.

molecular foot printing. Although we have not observed these transferred molecules at single-molecule resolution, which requires further development, molecules transferred on the flow cell surface from K562 cells were successfully visualized.

A decade has passed since the original smSBS emerged, and we believe that a revival of our system with specific improvements makes a significant contribution to not only spatial transcriptomics but also to spatial proteomics.

## Methods

**In-house sequencing system.** Our system was constructed using commercially available Nikon Ti-E (TI-ND6-PFS, Nikon) equipped with the following; ×60 objective lens (Apo TIRF ×60, Nikon) with ×1.5 transfer, EMCCD-camera (Ima-gEM-1K-N TDI-qEM-CCD, Hamamatsu Photonics), excitation laser 640 nm (CUBE 640–100 C, COHERENT), motorized stage system (BIOS-206T, SIGMA-KOKI) with our custom stage holder, a custom heater plate (Custom, Tokai hit) and a reagent exchanging system with eight syringe pumps (Cavro XLP6000 with 9-port valves and 1 ml syringe, TECAN); (Supplementary Fig. 12). A custom flow cell was placed on the stage and connected to the reagent exchange system via

PEEK tubes (Supplementary Fig. 13). Sequencing reagents were introduced through the reagent exchange system. To perform sequencing reactions, buffer supplies and image acquisitions were regulated using custom software written in C#.

**Flow cell construction with covalent bond capture probes**. Two holes, each with a diameter of 0.5 mm, were drilled into slide glasses (76 mm × 26 mm, No.1, S1111, Matsunami) with a diamond-tipped drill as shown (Supplementary Fig. 13). The slide glasses were then placed in a glass container filled with 100% acetone and sonicated for 30 min and rinsed with DI water. Subsequently, it was filled with 1 M KOH, sonicated for 30 min, and rinsed with DI water, following which the glasses were dried. A double-sided tape with a thickness of 30 μm (9313BT, 3 M) was cut and attached to the holed slide glass as shown (Supplementary Fig. 13). Then, an NHS-PEG cover glass (NHS_02, MicroSurfaces) was placed on top of the tape to create a channel with a volume of 1 μl. After the flow cell was mounted on the system, 10 μl of 1 μM of $NH_2$-$dT_{50}$ (5′-[AmC6]$T_{50}$–3′) or another capture oligo-nucleotide having a $NH_2$ group at the 5′ end in 0.3 M phosphate buffer (pH 8.5) was loaded using a fluidic pump from the sequencing system and incubated at room temperature (RT) for 1 h. Subsequently it was rinsed with 60 μl of 1× PBS twice. Unreacted NHS molecules were deactivated using 75 μl of deactivating buffer and incubating at RT for 30 min, after which the flow cell was rinsed with 75 μl of 1× PBS twice.

**Flow cell construction with non-covalent bond capture probes**. The flow cell was assembled using the same procedure for covalent bond capture probes with the modification of a biotin-PEG cover glass (Bio_02, MicroSurfaces) instead of NHS-PEG cover glass. After the flow cell was assembled with the biotin functionalized cover glass and mounted onto the system, 10 μl of 0.1 mg/ml of neutravidin in 1× PBS was loaded and incubated at RT for 5 min, following which the flow cell was rinsed with 75 μl of 1× PBS twice. Subsequently, 10 μl of 1 nM of biotin-$dT_{50}$ (5′-[BioON]$T_{50}$–3′) or biotin × 4-$dT_{50}$ (5′-[BioON]$T_{10}$[BioON]$T_{10}$[BioON]$T_{10}$[BioON]$T_{20}$–3′) was loaded and incubated at RT for 30 min. Next, the flow cell was rinsed twice with 75 μl of 1× PBS.

**Sample capture**. After attaching sample capture oligos to the flow cell, the temperature of the flow cell was set at 37 °C, and rinsed with hybridization buffer (1× SSC, 0.05% SDS). The sample, diluted to an appropriate concentration in hybridization buffer, was introduced and incubated at 37 °C for 1 h, after which the flow cell was rinsed with Wash A buffer (150 mM HEPES (KOH, pH 7.0), 1× SSC, 0.1% SDS) and Wash B buffer (150 mM HEPES (KOH, pH 7.0), 150 mM NaCl).

**Fill-and-lock step**. To sequence the sample captured via poly-A-$dT_{50}$ hybridization, the following fill-and-lock step is required. After capturing the sample, the flow cell was incubated with fill-and-lock buffer (20 mM Tris-HCl (pH 8.8), 10 mM KCl, 10 mM NaCl, 10 mM $(NH_4)_2SO_4$, 0.1% Triton X-100, 150 μM $MnSO_4$, 50 U/ml Klenow (exo-) and 1 μM dTTP, 200 nM VT-A, 125 nM VT-C, and 75 nM VT-G) at 37 °C for 4 min, following which the flow cell was rinsed with Wash A and Wash B three times. In the case of sample captured by a specific primer, fill-and-lock buffer was added with 125 nM VT-T and 2 min of incubation time instead of 1 μM dTTP and 4 min, respectively. We used virtual terminators supplied by Helicos or synthesized by Shinsei Chemical Company, Ltd. (Osaka, Japan). We outsourced synthesizing VTs to the chemical company according to the procedure described in the patent[25].

Subsequently, fluorescence images were captured with a 200-ms exposure with imaging buffer (100 mM HEPES, 67 mM NaCl, 25 mM MES, 12 mM Trolox, 5 mM DABCO, 80 mM glucose, 5 mM NaI, and 0.1 U/μL glucoseoxidase, pH 7.0) prepared immediately before being introduced to the flow cell. After imaging, the flow cell was rinsed with Wash A and Wash B. The position of molecules identified from images taken at the fill-and-lock step was considered as the "initial position". After imaging, cleave buffer (250 mM Tris-HCl, 100 mM NaCl, 50 mM TCEP-HCl, and pH 7.6) was incubated at 37 °C for 5 min to cleave fluorescence dye and the inhibition group bound to VTs. Iodoacetamide buffer (100 mM Tris-HCl, 100 mM NaCl, 50 mM iodoacetamide, and pH 9.0) was then added and the mixture was incubated at 37 °C for 5 min to deactivate exposed SH-groups. Following the cleaving step, the flow cell was rinsed with Wash A and Wash B, imaged again to confirm cleavage and rinsed with Wash A and Wash B. Sixty microliters of all buffers, except the fill-and-lock buffer, were introduced into the flow cell at a flow rate of 3 μl/s, while 10 μl of the fill-and-lock buffer was introduced at a flow rate of 3 μl/s.

**Sequence procedure**. After the fill-and-lock step, each sequencing cycle was performed as follows. The flow cell was incubated at 37 °C for 2 min with VT incorporation buffer (20 mM Tris-HCl (pH 8.8), 10 mM KCl, 10 mM NaCl, 10 mM $(NH_4)_2SO_4$, 0.1% Triton X-100, 5 U/ml Therminator™ DNA polymerase (NEB) and 1 mM $MgSO_4$ with either 125 nM VT-C, 125 nM VT-T, 200 nM VT-A, or 75 nM VT-G), followed by rinsing with Wash A and Wash B. Next, the flow cell was filled with freshly prepared imaging buffer, and fluorescence images were captured with a 200-ms exposure, after which the flow cell was rinsed with Wash A and Wash B. Subsequently, cleavage buffer and iodoacetamide buffer were respectively incubated

at 37 °C for 5 min, and the flow cell was rinsed with Wash A and Wash B. Again, the imaging process, followed by washing with Wash A and Wash B was performed.

**Base calling (image analysis, stage drift compensation)**. First, the positions of individual VT incorporations in each FOV were identified as pixel coordinates in integer values with software customized by Hamamatsu photonics[43]. Next, we performed two rounds of the stage drift correction process. Note, we conducted this based on the pixel coordinates of individual VT incorporations identified via the software, not by applying a direct image correction process (such as cross-correlation). We expected that individual VT incorporations at each cycle would be overlapped to the corresponding position at the 1st cycle, where all molecules attempt to incorporate VTs. Thus, in the first round, the correction (translation) value of individual FOVs was determined so that the translated FOV shows maximum matching of molecules to those corresponding in the 1st cycle. Next, as the position markers, we extracted molecules at the 1st cycle showing the top 10% frequently observed (matched to) VT incorporations. In the second round, the correction (translation) value of individual FOVs was again determined so that the translated FOV indicated the maximum matching of molecules to "the position markers". Following drift compensation, individual VT incorporations matched to initial positions were identified with a tolerance of one pixel and transformed into sequence information (base calling). Reads that failed to cleave were excluded by examining the images after cleavage.

**Sample preparation (in-house barcode molecules)**. Thirty types of poly-A tailed DNA barcode molecules were prepared by referring to the sequence of Helicos control oligo. Firstly, we added a poly-A tail to the oligonucleotides (Supplementary Table 1) by incubating 10 μl of mixture consisting of 10 μM oligonucleotide mix, 1× TdT buffer, 250 μM $CoCl_2$, 240 μM dATP, and 1 U/μl terminal transferase at 37 °C for 1 h and 70 °C for 10 min followed by a 4-°C hold. Secondly, the sample was added to 10 μl of mixture consisting of 1× TdT buffer, 250 μM $CoCl_2$, 160 μM biotin-11-ddATP (PerkinElmer), and 2 U/μl terminal transferase (NEB) and incubated at 37 °C for 1 h, and 70 °C for 10 min followed by a 4-°C hold.

**Barcode identification**. We mapped the sequence reads of 30 types of barcode molecules to the barcode ID reference (Supplementary Table 1) using BLAST (2.2.30+) with the following option (blastn -db [barcode.reference] -query [barcode_read.fasta] -out [out.file] -word_size 4 -num_descriptions 1 -num_alignments 1 -dust no -strand plus -outfmt "7 std qlen qseq sseq"), and identified barcode IDs with BLAST E-values < 0.01.

**cDNA sequencing with smSBS (K562 full-length cDNA)**. We lysed 30 K562 cells using ×10 Reaction Buffer from a SMART-seq v4 kit (Clontech) following the manufacturer's protocol. Next, 1st strand synthesis was performed via SMART-Scribe Reverse Transcriptase (Clontech) with an RT primer (5′-AAGCAGTGGTATCAACGCAGAGTAC($dT_{30}$)VN-3′) and a TSO (5′-AAGCAGTGGTATCAACGCAGAGTACrGrG+G-3′) at 42 °C for 90 min. Subsequently, this mixture was divided into four aliquots, one of which was measured with and without PCR amplification, respectively. In the case of the amplification-free aliquot, further ExoI treatment was performed following 1st strand synthesis. For the amplified sample, cDNA was amplified via PCR (95 °C for 1 min; 18 cycles of 98 °C for 10 s; 65 °C for 30 s; 68 °C for 3 min; and 72 °C for 10 min) with SeqAmp DNA polymerase from the SMART-seq v4 kit and a PCR primer (5′-AAGCAGTGGTAT-CAACGCAGAGT-3′), followed by purification with AMPure XP (Beckman Coulter). The yield and quality of cDNA were estimated with a Qubit 2.0 Fluorometer (Thermo Fisher Scientific). The cDNA library was diluted and sequenced on our smSBS with 24Q sequence cycles.

**cDNA sequencing with Illumina-seq**. For this purpose, 200 pg of total RNA purified from bulk K562 cells were converted to cDNA using the same procedure described above, with 15 PCR cycles applied. Additionally, cDNA molecules were fragmented using a Nextera XT DNA sample preparation kit (Illumina) and purified according to the manufacturer's protocol, followed by sequencing on an Illumina HiSeq 2500 with 100-base paired-end reads.

**Data analysis for smSBS data (mapping and gene counting)**. Sequence reads <5 nt, or matched to byproduct reads, were filtered out. Byproduct reads were identified by mapping them to the byproduct references as shown (Supplementary Table 2). Pre-filtered reads of smSBS were mapped to the human reference (GRCh37.75) using STAR[44] (version 2.5.1b) with ENCODE standard options with the exception of (−outFilterMismatchNoverLmax 0.3−outFilterScoreMinOverLread 0.3−outFilterMatchNminOverLread 0.3), and the number of genes was counted with HTseq-count[45] (version 0.11.2) using Homo_sapiens.GRCh37.75.gtf with an option (-a0 -s no -m intersection-nonempty−nonunique all−secondary-alignments score).

**Data analysis for Illumina data (mapping and gene counting).** We mapped the trimmed sequencing reads to the human reference (GRCh37.75) using the STAR (version 2.5.1b) mapping program with ENCODE options, and calculated expression estimates with TPM using RSEM (v1.3.0)[46].

**DNA barcode tagged antibody measurement.** First, a complementary strand of the DNA barcode molecule was synthesized with the biotin-dT$_{50}$ as a primer in a general PCR tube (Supplementary Fig. 9a). Next, we transferred double-stranded DNA barcode tagging antibodies to a flow cell (Fig. 4a, leftmost), and the templated strand was dissociated by denaturing with 50% formamide. Subsequently, the complementary strand of the barcode leaving the flow cell was re-hybridized with a sequencing primer and sequenced (Fig. 4a, rightmost). In detail, we synthesized a complementary strand of DNA barcode molecules bound to the antibody by incubating a mixture consisting of 1× NEB2 buffer, 1 nM AB-oligo (TotalSeq™-A0383 anti-human CD55 Antibody, Biolegend), 10 nM biotin ×4-dT$_{50}$, 50 μM dNTPs, and a 50-U/μl Klenow Fragment (NEB) at 37 °C for 15 min. We diluted these 10-fold with 1× PBS, and loaded 4 μl into a biotin-avidin flow cell followed by 25 min incubation at RT. The flow cell was rinsed with 75 μl of 1× PBS twice, incubated with 50% formamide at 50 °C for 5 min and rinsed with 1× PBS twice, and 75 μl of pre-hybridization buffer. Next, 0.5 μM of PCR handle primer (5′-CCTTGGCACCCGAGAATTCC-3′) was loaded onto the flow cell and incubated at 50 °C for 1 h. The flow cell was then rinsed with Wash A and Wash B and its temperature was lowered to 37 °C. Next, we performed the fill-and-lock step, followed by 12Q sequencing.

**Molecular foot printing.** First, 0.25 μl of 1× Ab-oligo solution (TotalSeq™-A0361 anti-human CD59 antibody, Biolegend), 0.25 μl of 100 μM A0361 complementary sequences (5′-[BioON]T$_{10}$[BioON]T$_{10}$[BioON]T$_{10}$[BioON] T$_{20}$VTCTCGACGGC-TAATTTGGAATTCTCGGGTGCCAAGG-3′) and 22 μl of 1 × PBS were mixed and incubated at 65 °C for 5 min, followed by on-ice incubation for 2 min. Next, 2.5 μl of 1× protein-G coated magnetic beads (NEB, S1430S) was added to the mixture and incubate at RT for 1 h with gentle mixing by a rotator. The beads were then washed with 25 μl PBS four times using a magnetic stand.

The flow cell was assembled using a biotin-PEG cover glass (Bio_02, MicroSurfaces), after which the biotin functionalized cover glass was mounted onto the system, and 10 μl of 0.1 mg/ml of neutravidin in 1× PBS was loaded and incubated at RT for 5 min, following which the flow cell was rinsed with 75 μl of 1× PBS twice.

The protein-G coated beads solution was introduced into the flow cell and captured on the surface. After observing the beads, they were washed from the flow cell via a pressure driven flow of 50% formamide solution. The complementary sequences remaining on the surface were visualized by incorporating VTs as described previously in "DNA barcode tagged antibody measurement".

When K562 cells were applied to the system, the 5′-[BioON]-dT$_{50}$- Cy5-3′ probe was used in place of the complementary sequences. First, 1 ml of ~10$^5$ cells/ml of K562 cells was washed with a staining buffer (1× PBS containing 2% BSA and 0.01% Tween20) by centrifugation (350×g, 4 min, 4 °C), followed by resuspension with 100 μl of staining buffer. Then, 0.25 μl of 1× Ab-oligo solution (TotalSeq™-A0383 anti-human CD55 antibody, Biolegend) and 0.25 μl of 100 μM 5′-[BioON]-dT$_{50}$- Cy5-3′ probe were added into the cell suspension and incubated for 30 min on-ice. The cell mixture was then washed with 1 ml staining buffer four times by centrifugation (350 × g, 5 min, 4 °C), and resuspended with 100 μl of staining buffer. Transfer of the 5′-[BioON]-dT$_{50}$- Cy5-3′ probe to the K562 cells onto a sequencing flow cell was performed in the same way as that described for the protein-G coated bead.

**Statistics and reproducibility.** Statistical analyses were conducted using R version 3.5.1. Difference in read length between groups was examined by a Mann–Whitney $U$ test. All experiments, except K562 cDNA library sequencing, were repeated at least twice. The number of FOVs for individual experiments for barcode identification is in the corresponding figure legend.

**Reporting summary.** Further information on research design is available in the Nature Research Reporting Summary linked to this article.

## Data availability
The sequencing data from this study have been deposited in the DDBJ DRA database under the accession number DRA009022. The raw data referring to the plots shown in the main figures are provided in Supplementary Data 1. All relevant data are available from the authors upon request.

## Code availability
The custom code regulating our sequencing system can be purchased from NIKON SOLUTIONS CO., LTD.; the code is integrated with a commercially available software (Hamamatsu photonics) and works only with the system described in the methods section.

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

## Acknowledgements

The authors would like to thank Dr. Yoshihide Hayashizaki for his scientific advice and Shinsei Chemical Company, Ltd. (Osaka, Japan) for synthesizing virtual terminator nucleotides. This study was supported by the ImPACT Program of the Council for Science, Technology, and Innovation (Cabinet Office, Government of Japan), Japan Society for the Promotion of Science (grant no. 18K06195) and JST, PRESTO, Japan (grant no. JPMJPR1943).

## Author contributions

Y.O. developed the sequencing system and performed the experiments. H.S. prepared the samples for sequencing. Y.O. analyzed the data. All authors wrote the manuscript. Y.O. and S.U. designed and supervised the study.

## Competing interests

The authors declare the following competing interests: Y.O. and S.U. have filed a Japanese patent (no. 6288650) for the flow cell described in the current study. All other authors declare no competing interests.
