## [Peer Review File · Communications Biology]

Reviewers' comments:

Reviewer #1 (Remarks to the Author):

Although authors were able to recapitulate sequencing concept of Helicos BioSciences I was struggling to understand and envision what advantage it will provide to the scientific community. First, sequencing was achieved only on relatively short DNA fragments (oligonucleotides below 50 nt long), setting up the in-house system will be a challenging task for most laboratories. Sequencing DNA libraries these days is so cheap that setting up your own sequencer, with higher error rates than commercial system will be disfavored by majority of groups. Furthermore, the base calling is not trivial process and requires customized software that is not readily available. I admire authors efforts to be honest and report imperfections and weaknesses of the system. This will be helpful in the future for further developments and improvements, but I think at this stage this work lacks conceptual novelty and is insignificant in terms of technological advance. Unfortunately I cannot recommend this work for publication in this journal.

Molecular footprinting is an interesting idea but the proof-of-concept reported here is insufficient for publication at this journal. It simply lacks sufficient information and it seems a result of selected experiment. The CITE-Seq barcodes are captured on one side of the cell, the percentage of Ab-DNA that were extended by RT is not clear. I would expect that efficiency of RT reaction in the reported conditions will be around 10%. It is not clear how Ab::epitope interaction can be maintained during RT reaction. Ab would tend to denature at temperatures higher than 37°C, while RT needs at least 42°C. How many CITE-Seq barcodes/molecules are lost during RT?

In many parts of the manuscript authors claim single-molecule sensitivity. While I agree that system provides single-molecule sensitivity, this is not something superior to any other sequencing platform. What is crucial however is sensitivity and error-rate. For example, if one aims to sequence e.g. 10,000 unique cDNA molecules how many will be lost in the process?

Few other points that I hope will help you to improve the work:

- Please provide the estimate of cost of sequencing.
- What constitutes the "deactivating buffer" ?
- It was not clearly presented what is error rate when there are two subsequent nucleotides are of identical chemical nature e.g. AA, CC, GG etc.
- I believe this statement is inappropriate: "... our system ... offers unique gene detection of full-length cDNA molecules without fragmentation" The system is not able to detect full-length cDNA molecules because it can sequence only up to 50 bases. Authors should rephrase it.
- I would not say that this work "reconstructed smSBS including sequencing chemistries, independently of Helicos BioScience". Authors have used the buffers that they found in Helicos patents which means that authors depend on Helicos BioScience previous work.
- Authors should cite previous work: doi: 10.1002/0471142727.mb0710s92

Development of in-house sequencing system towards spatial-decoding of DNA barcode molecules at single-molecule resolution

Yusuke Oguchi *et al.* devised an in house system that mimics the abilities of HeliScope, the single molecule high-throughput DNA sequencer. The system was tested for several different conditions: (i) Conventional sequencing of 30 predesigned DNA sequences (barcodes) (ii) cDNA sequencing through direct hybridization of the template switching oligo (TSO) to a complementary oligo, capture probe, on the surface (iii) Quantification of the effect of covalent vs. two non-covalent attachment chemistries of the capture probes on the performance of the sequencing process (iv) Biotin-Avidin attachment and sequencing of oligoes complementary to DNA barcode of antibody.

Over all, this work represents an impressive endeavor and should be of interest to the community. However, some points need to be better addressed:

1. The authors don't actually show 'spatial-decoding of DNA barcode molecules'. Therefore, the title and abstract should be rephrased to state the main findings and not a desired future application.
2. A major limitation of single-molecule sequencing is read length. It would be informative if the authors show the length distribution of their barcode sequencing (in a histogram), as well as for their additional sequencing experiments. Second, as both barcode and cDNA sequencing take a significant amount of the read length, how do the author envision they can be combined in future applications?
1. Also related to length: The authors state (page 6) that 73% of their reads aligned with the human genome. They also note that they considered every read longer than 4 nucleotides. What is the meaning of alignment of such short reads? A better approach would be to report how many reads they obtained **longer than 18bp**, and then how many of these aligned to the genome.

Minor comments

1. Add a summary of the main sequencing errors with respect to the number of insertions, deletion and misincorporations. Are there any sequence dependent biases?
2. Fig. 1A – The picture is not self-explanatory, denote the different parts of the system.
3. Fig. 1B – The illustration is ambiguous. It does not contribute to better understanding of the process. Add a more detailed explanation of the single molecule sequencing process or remove it.
4. The synthesis of virtual terminators should be mentioned in the Methods section.
5. What were the considerations in the barcode design?
6. What were the BLAST parameters used for alignment?
7. Lines 182-184 – Elaborate more on the difference between the enzymes in the main text.

Reviewer #3 (Remarks to the Author):

In this work, Oguchi et al. propose restructuring existing sequencing method (HeliScope) to instead perform spatial resolution of DNA barcodes at single-molecule level. To demonstrate their approach, they first read 20 nt DNA sequences to spatially resolve 30 different DNA barcodes. They also demonstrate their approach in gene quantification context by trying to spatially resolve cDNA library for K62 cell. They also propose replacing prior flow cell binding chemistry with biotin-avidin which is a standard in microscopy. Finally, as a real-world application, they try to spatially identify DNA barcodes bound to antibodies to demonstrate that their method can be applied at the protein-level.

Usually there is a space v/s counts tradeoff as the authors correctly identify. If one desires to identify more types of barcodes, they can't be too close spatially and vice versa. The authors argue that since industry tool such as sequencers are proprietary, not much information is available on how they work. Therefore, to bridge the gap, they break apart Heliscope, which is used routinely for DNA/RNA sequencing at scale, and instead re-purpose it to do single-molecule barcoding.

I like the overall idea in this work! Currently there are fancy barcoding techniques at single-molecule level but they all have their challenges. For example, geometric barcoding requires extensive preparation, temporal barcoding requires only one dye but longer data collection times, washing based techniques introduce additional drift in the sample, etc. Instead of trying to come up with a completely new technique to do single-molecule barcoding, why not simply use existing old sequencing machine lying in the lab since often times it is very difficult to setup a new barcoding technique in lab. Obviously then the question is how good will their method be? In this case, the authors show that it is reasonably versatile since they can do up to 30 barcodes with high accuracy and speed however they were all short strands (< 20 nt).

Here are my comments:

1. Based on figure 1D, it seems like a lot of overlap between the individual barcodes in the fluorescence image. If there were two strands present in the same spot (or nearby), without super resolution imaging, it is impossible to determine whether the fluorescent blink came from which strand. Other studies such as DNA origami barcodes or temporal DNA barcodes have used concentration as low as 20 pM to avoid spatial overlap since biotin-avidin locations are non-specific on glass surface. The experiments sections say authors operated at 10 uM. The authors need to perform their sequencing experiments at a lot lower concentration. Even then they can't guarantee that more than one strands one sit in the same diffraction limited spot but the likelihood is a lot lower. Refer to these studies for the experimental protocol to minimize the spatial overlap of fluorescent spots and cite them appropriately.

a. Lin, Chenxiang, et al. "Submicrometre geometrically encoded fluorescent barcodes self-assembled from DNA." *Nature chemistry* 4.10 (2012): 832-839.

b. Shah, Shalin, Abhishek K. Dubey, and John Reif. "Programming temporal DNA barcodes for single-molecule fingerprinting." *Nano letters* 19.4 (2019): 2668-2673.

2. How exactly do authors do drift-correction is not very clear. They authors mention that they use Hatamatsu software but what algorithm do they use? Redundant cross-correlation, cross-correlation, Direct cross-correlation? One standard trick in imaging is to incubate sample on microscope stage for a few minutes before imaging starts to minimize thermal drift. I see that the authors write 2 – 4 minutes of incubation. It doesn't seem enough to me. Prior works have suggested 10 – 20 minutes of incubation time. Some imaging protocols:

a. Roy, Rahul, Sungchul Hohng, and Taekjip Ha. "A practical guide to single-molecule FRET." *Nature methods* 5.6 (2008): 507-516.

b. Schnitzbauer, Joerg, et al. "Super-resolution microscopy with DNA-PAINT." *Nature protocols* 12.6 (2017): 1198.

3. The authors in this work are trying to do single-molecule imaging with DNA to barcode (or uniquely identify them). However, I see no mention to the recent temporal DNA barcodes work or geometric barcodes work or washing based techniques etc. They should cite all the related references and add a small comparison (may be a table or paragraph) since the current work and prior works have same end-goal but are used in different contexts (fluorescence microscopy vs sequencing-by-imaging). For example, temporal barcode benefit is it uses only one dye, geometric barcode benefit is it can do 100s of barcodes, washing technique is useful because it can program 4^N barcodes, etc. While I see that the authors briefly talked about FISHseq+ in discussion that's only scratching surface. There a full field of DNA-based barcoding out there that is missed in the references section (mostly sequencing references). The authors should also look at these other works and cite them as they will find that all these works have the same eventual goal (single-molecule barcoding).

a. Lin, Chenxiang, et al. "Submicrometre geometrically encoded fluorescent barcodes self-assembled from DNA." *Nature chemistry* 4.10 (2012): 832-839.

b. Johnson-Buck, Alexander, et al. "Kinetic fingerprinting to identify and count single nucleic acids." *Nature biotechnology* 33.7 (2015): 730-732.

c. Jungmann, Ralf, et al. "Multiplexed 3D cellular super-resolution imaging with DNA-PAINT and Exchange-PAINT." *Nature methods* 11.3 (2014): 313-318.

d. Shah, Shalin, Abhishek K. Dubey, and John Reif. "Programming temporal DNA barcodes for single-molecule fingerprinting." *Nano letters* 19.4 (2019): 2668-2673.

e. Shah, Shalin, Abhishek K. Dubey, and John Reif. "Improved optical multiplexing with temporal DNA barcodes." *ACS Synthetic Biology* 8.5 (2019): 1100-1111.

4. Similar to drift-correction, I see that the flow chamber used by authors have 1x PBS buffer, other redox systems, and salts in their wash buffer. Usually it is not enough to avoid non-specific binding that occurs due to dye sticking on the surface and giving a false-positive. Did they do some treatment of glass surface before setting up flow chamber? Usually people use PEG for avoiding non-specific binding.

5. Have the authors tried different capture times? I see that they use 200 ms exposure times, how did they come up with this since they will lose events faster than 200 ms and supposed if a base gets synthesized in 50 ms then they won't capture it in their read. Some explanation regarding this is required.

6. The authors on page 4 line 102 say that per quad they needed about 2 hours of data collection time and in total 2 days. This seems pretty slow. I was hoping that since they have multi-color dye and existing sequencer this should happen a lot faster. Can they talk about how they decided on the amount of data collection time?

7. On page 5 line 112 the authors argue the readable length of a barcode increases with the increase in the number of sequencing cycles. While it sounds intuitively accurate, the figure 1f plot shows something else. It seems as if based on the average plot it nearly saturates after 20 nt i.e. at this point even if the data collection time is increased, one cannot read longer DNA. Is that correct? Can the authors comment on this? If so, they should talk about it deliberately in their discussion section as this is an important trade off. Since the total number of barcodes are 4^N where N is the length of barcode, learning how to accurately identify longer barcodes will be useful.

8. On page 9 line 222, the authors argue that overall by incorporating their technique the number of barcodes will scale up to 200. Can they please provide a reasoning to this? Some equation or calculation or justification on how they came up with this number since they show upto 30 in the paper and for figure 4g the efficiency is just slightly above 50%.

Response to Reviewers:

Reviewer #1 (Remarks to the Author):

Although authors were able to recapitulate sequencing concept of Helicos BioSciences I was struggling to understand and envision what advantage it will provide to the scientific community. First, sequencing was achieved only on relatively short DNA fragments (oligonucleotides below 50 nt long), setting up the in-house system will be a challenging task for most laboratories. Sequencing DNA libraries these days is so cheap that setting up your own sequencer, with higher error rates than commercial system will be disfavored by majority of groups. Furthermore, the base calling is not trivial process and requires customized software that is not readily available. I admire authors efforts to be honest and report imperfections and weaknesses of the system. This will be helpful in the future for further developments and improvements, but I think at this stage this work lacks conceptual novelty and is insignificant in terms of technological advance. Unfortunately I cannot recommend this work for publication in this journal.

We appreciate your comments. We agree that our original manuscript may have lacked sufficient data to support conceptual novelty and technological advance, therefore, we have performed two additional experiments to strengthen the study (as new Fig.5 and Fig.S10). We believe that our novelty must be supported by additional data, which will pave the way for achieving spatial transcriptomics as well as spatial proteomics at the single molecule level, thereby making a significant contribution to the scientific community in the future.

Molecular footprinting is an interesting idea but the proof-of-concept reported here is insufficient for publication at this journal. It simply lacks sufficient information and it seems a result of selected experiment. The CITE-Seq barcodes are captured on one side of the cell, the percentage of Ab-DNA that were extended by RT is not clear. I would expect that efficiency of RT reaction in the reported conditions will be around 10%. It is not clear how Ab::epitope interaction can be maintained during RT reaction. Ab would tend to denature at temperatures higher than 37°C, while RT needs at least 42°C. How many CITE-Seq barcodes/molecules are lost during RT?

First, we appreciate the insightful attention to the CITE-Seq barcodes, which is recently widely accepted by the community for achieving simultaneous analyses of transcriptome and proteome data with the same single-cells. As reviewer #1 pointed out, it is important how efficiently maintaining Ab::epitope complex to be workable during RT reaction, the efficiency of this would be impacted by many factors, such as extension efficiency and Ab denaturation by high temperatures. Here we focused on examining the temperature effect by hybridizing the pre-synthesized complementary strand to the barcode sequence (not requiring the extension). Additionally, we conducted a hybridization of the Ab::epitope complex and the complementary strand at 65 °C, which is expected to be a harsher condition than that of standard RT reaction. Thus, if we can obtain proof-of-concept data with this complex, it would be helpful to reduce reviewer#1's concern regarding Ab denaturation by high temperatures.

Accordingly, as a proof-of-concept for molecular footprinting, first, we used protein-G coated beads mimicking cells (Figure 1 in this response document and Fig. 5 in the revised manuscript). The beads binding the Ab::epitope complex, prepared as mentioned above, were introduced into a sequencing flow cell. We confirmed that the complementary strands of barcode molecules were successfully transferred onto the flow cell surface from the beads, while the beads were washed from the sequence flow cell. These transferred molecules on the surface were densely sited reflecting the bead shape, which were visualized by virtual terminator incorporation as saturating fluorescence spots (fluorescence image). Further, we conducted sequence cycles with this sample and observed a correct order of VT incorporations to these spots according to the barcode sequence (Sequential images). Thus, we were able to transform DNA molecules binding an object to a sequencing flow cell, and showed that these transformed DNA molecules were then able to be sequenced with our system.

Fig 1. A proof of concept experiment for molecular foot printing. a) Ab-barcode complex with a pre-synthesized oligonucleotide of the complementary strand. Barcode-strand was double-stranded with an oligonucleotide of the complementary sequence of the barcode instead of reverse transcription. b) Schematic illustration of the experiment. Protein-G coated magnetic beads binding with the double-stranded Ab-barcode were introduced into a sequencing flow cell and captured on the surface via biotin-avidin interactions (left). After observing the beads, they were washed from the flow cell via a pressure driven flow of 50% formamide solution, while the complementary sequences remained on the surface (middle). Visualization of the complementary sequences remained on the surface by incorporating VTs (right). c) Microscopic images corresponding to the schematic illustration shown in b. Bright field images of protein-G coated beads on the surface (encircled in red, left panel). After washing, the beads were confirmed to have moved from the original position (middle). Fluorescence image of the complementary sequences remained on the surface visualized by VT incorporations

(right panel). Scale bars shown in c indicate 5 μm . d) Overlaid images of each cycle (green) on the image of the 1st cycle (magenta). Rectangles in orange indicate expected cycles of VT incorporation according to the barcode sequence. Sequencing proceeded from left to right (CTAG) and top to bottom (1Q to 3Q) in this sequential image.

Next, we performed a similar experiment with K562 cells in place of beads (Figure 2 in this document and Figure S10 in the revised manuscript). Note that, as the complementary strand of the barcode sequence, here we used a synthesized oligonucleotide (5'-Biotin-dT₅₀-Cy5-3') that binds to only the poly-A region of the barcode sequence. Also, the Cy5 fluorophore labeled in the probe allowed us to easily identify whether these strands were transferred onto the flow cell surface without sequencing. As mentioned above, these probes were pre-hybridized before adding to the cells, followed by mixing with the cells. After 1 h incubation at RT (room temperature), unbound probes were washed out via centrifugation (300 \times g for 3 min) three times. The labeled K562 cells were introduced into a sequencing flow cell and captured on the surface via biotin-avidin interactions (Figures in left). After observing the cells via bright field and fluorescence imaging, the cells were washed off the flow cell via a pressure-driven flow. We confirmed that the cells were washed away via bright filed imaging (at least, the cells were dissociated and relocated away from their original positions), while the fluorescence probes remained at the same position. Thus, we were able to show transformation of the molecular distribution of CD55 on K562 cells onto a sequencing flow cell surface.

Figure 2. A proof-of-concept experiment for molecular foot-printing with K562 cells. a) Instead of the complementary sequence shown Figure 1a, here, 5'-Biotin-dT₅₀- Cy5-3' probe was just hybridized without a further extension reaction with polymerases. b) Schematic illustration of the experiment with K562 cells. K562 cells bound Ab::epitope were introduced into a flow cell and captured onto the surface via biotin-avidin interactions (left panel). After observing the beads capturing on the surface, these beads were washed away from the flow cell via a pressure driven flow, while the complementary sequences were remained on the surface and easily visualized by Cy5 molecules (right). c) Bright field images corresponding to the schematic shown in b. d) Fluorescence (TIRF) images corresponding to the schematic shown in b. After confirming the K562 cells were washed away from the original position, the same FOV was observed via a fluorescence imaging. Scale bars shown in b and c indicate 5 μ m.

Taken together the data from these two additional experiments provided proof of concept for molecular footprinting. Although we have not observed these transferred molecules at single-molecule resolution, which requires further development, molecules transferred on the surface from K562 cells might show heterogenous distributions, and thus we believe that these data are sufficient for supporting our conceptual novelties contributing to spatial analyses of the transcriptome and proteome. We have provided these two additional experiments in the revised Results section,

In many parts of the manuscript authors claim single-molecule sensitivity. While I agree that system provides single-molecule sensitivity, this is not something superior to any other sequencing platform. What is crucial however is sensitivity and error-rate. For example, if one aims to sequence e.g. 10,000 unique cDNA molecules how many will be lost in the process?

We appreciate the comment. To estimate how many molecules will be lost if we aim to identify 10,000 unique molecules, we performed a simulation analysis based on the sequencing accuracy. We are able to reproduce the experimental results of the identification efficiency of 30 types of barcodes by a simple model analysis (Fig.3a-d in this manuscript, although the sequencing accuracy estimated by experiments was ~0.95, this model gives a good fitting with ~0.85). According to this model, the identification efficiency of 10,000 unique molecules can be reached to ~50% with 20 bp read length (Fig.3e in this manuscript). Thus, when we attempt to identify 10,000 unique molecules, ~5,000 molecules would be lost by identifying with ~20 bp read length. If we are able to increase the read length to ~30 bp on average, ~8,000 molecules should be identified. We provided a schematic in Fig. S11 and mentioned this in the discussion as follows:

Although, in the current study we demonstrated that our system effectively multiplexes up to 30 targets, we were able to further estimate the multiplexing capacity using a simple model (Fig. S11a-d), which indicated that our system would be potentially applicable for up to 10,000 unique molecules (Fig. S11e& f).

Figure 3. A model analysis estimating the identification of 10,000 unique molecules. a-d) Validation of model analysis with the experimental identification of 30 barcode molecules. a) Procedure of creating mock sequence reads for this simulation. First, 10,000 barcode molecules were randomly sampled from the 30 barcode molecules with replacement, followed by error insertion to mimic the experimentally obtained sequence reads. b) The fraction of reads containing n errors ($P(n)$) should be governed by a binomial distribution ($P(n) = {}_N C_n (1-p)^n p^{(N-n)}$, where p : sequencing accuracy, N : read length, n : number of errors per read). Based on the distribution, we created mock sequencing reads by inserting errors (here we simply inserted substitution errors) into the original sequences. c) Identification of mock sequencing reads by aligning them to the original 30 barcode sequences with BLAST. d) Comparison of simulated identification efficiencies with the experimentally obtained result. The result of simulation with $p = 0.85$ provides a good fit for the experiment. Thus, we estimate an identification efficiency of 10,000 unique molecules with $p = 0.85$ and ranging from 15 to 30 bp in the read length (e). Analysis with $e\text{-value} < 0.01$ shows almost no incorrect identifications, therefore, a looser threshold, such as $e\text{-value} < 0.1$, can be applied, resulting in increased identification efficiency, especially with 15 bp read length (f). Definitions for identification of molecules are described in c.

Few other points that I hope will help you to improve the work:

- Please provide the estimate of cost of sequencing.

~\$250 per run (24Q) for sequencing consumables including flow cell, reagents, virtual terminators and DNA polymerases, and ~\$150,000 for constructing a system.

- What constitutes the "deactivating buffer"?

Although the deactivating buffer was provided by the supplier of the NHS-PEG coverslips (Microsurfaces), and the actual composition of this buffer is not disclosed, it would contain an amine-group (such as Tris) to "deactivate" NHS groups that remained unreacted after the coupling reaction.

- It was not clearly presented what is error rate when there are two subsequent nucleotides are of identical chemical nature e.g. AA, CC, GG etc.

Thank you for the insightful comment. We added further analysis of the error rates (Fig.4 in this document and Figure S3 in the revised manuscript). We carefully re-analyzed the error rate of the 2nd incorporation after the 1st between two subsequent incorporations (and comparing two technical replicates, ThI_1: Terminator polymerase replicate 1, ThI_2: Terminator polymerase replicate 2). We found the error rate of the 2nd incorporation in CC, GG, and CG to be slightly higher (~10%) than the others, while the others are less than ~5%. In addition, the composition of errors is shown in the right two panels below, the error of our system is primarily attributed to the deletion and insertion errors, similar to that previously reported by Helicos. We added this as the New Fig.S3b-d, combining with the original Fig.S2g.

Figure 4. Error rate of the second incorporation of two subsequent incorporations. a) Comparison in the error rate of the second incorporation between technical replicates (rep.1 vs. rep.2). For instance, CG indicates the error rate of G incorporation after the C incorporation. Fraction of error types by each subsequent incorporation for technical replicates 1 (c) and 2 (d).

- I believe this statement is inappropriate: "... our system ... offers unique gene detection of full-length cDNA molecules without fragmentation" The system is not able to detect full-length cDNA molecules because it can sequence only up to 50 bases. Authors should rephrase it.

Thank you for the comment, we agree that our system is not able to obtain sequence coverage for the whole region of cDNA molecules. However, in contrast to recent Illumina sequencing methods that require a fragmentation step, our system offers a unique detection system using full-length cDNA molecules. That is, our system does not require the fragmentation of cDNA molecules, which might produce sequencing biases (ref.1-4 below). Therefore, we rephrased the text as follows:

Thus, our system is applicable to ~10 pg of cDNA input and offers unique gene detection of full-length cDNA molecules without fragmentation by detecting either the 5' or 3' end region, which may contribute to eliminating sequencing biases produced by the fragmentation process (ref.1-4).

1. Sato, M. P. *et al.* Comparison of the sequencing bias of currently available library preparation kits for Illumina sequencing of bacterial genomes and metagenomes. *DNA Res.* **26**, 391–398 (2019).
2. Green, B., Bouchier, C., Fairhead, C., Craig, N. L. & Cormack, B. P. Insertion site preference of Mu, Tn5, and Tn7 transposons. *Mob. DNA* **3**, 1–6 (2012).
3. Lan, J. H. *et al.* Impact of three Illumina library construction methods on GC bias and HLA genotype calling. *Hum. Immunol.* **76**, 166–75 (2015).
4. Kia, A. *et al.* Improved genome sequencing using an engineered transposase. *BMC Biotechnol.* **17**, 1–10 (2017).

- I would not say that this work "reconstructed smSBS including sequencing chemistries, independently of Helicos BioScience". Authors have used the buffers that they found in Helicos patents which means that authors depend on Helicos BioScience previous work.

We appreciate your correction. We rephrased this sentence in the discussion as follows:

In summary, the system described herein not only reconstructed smSBS by applying sequencing chemistries inspired by the original Helicos BioScience system, but also introduced further improvements such as barcode decoding, small quantity and full-length cDNA capturing and sequencing, as well as biotin-based durable sample capturing.

- Authors should cite previous work: doi: 10.1002/0471142727.mb0710s92

We cited the following study in our revised manuscript:

HeliScope, which was originally developed and commercialized by Helicos BioSciences, offers unbiased DNA sequencing^{11,17} and direct RNA sequencing¹⁸, and was leveraged for large-scale research¹⁹ by the international consortium, FANTOM.

17. Thompson, J. F. & Steinmann, K. E. Single molecule sequencing with a HeliScope genetic analysis system. *Curr. Protoc. Mol. Biol.* Chapter 7, Unit7.10 (2010).

Reviewer #2 (Remarks to the Author):

Yusuke Oguchi *et al.* devised an in house system that mimics the abilities of HeliScope, the single molecule high-throughput DNA sequencer. The system was tested for several different conditions: (i) Conventional sequencing of 30 predesigned DNA sequences (barcodes) (ii) cDNA sequencing through direct hybridization of the template switching oligo (TSO) to a complementary oligo, capture probe, on the surface (iii) Quantification of the effect of covalent vs. two non-covalent attachment chemistries of the capture probes on the performance of the sequencing process (iv) Biotin-Avidin attachment and sequencing of oligoes complementary to DNA barcode of antibody.

Over all, this work represents an impressive endeavor and should be of interest to the community. However, some points need to be better addressed:

We appreciate your comments.

1. The authors don't actually show 'spatial-decoding of DNA barcode molecules'. Therefore, the title and abstract should be rephrased to state the main findings and not a desired future application.

Thank you for the comment. We agree that our original manuscript may have lacked data that clearly demonstrated our contribution to spatial analyses, as was noted by reviewer #2 as well as reviewer #1. Thus, we performed an additional two experiments to strengthen the study. We would appreciate it if you would re-evaluate this with the additionally added results (please refer to Figs.1 & 2 in this manuscript and our responses to the reviewer #1 regarding these experiments).

2. A major limitation of single-molecule sequencing is read length. It would be informative if the authors show the length distribution of their barcode sequencing (in a histogram), as well as for their additional sequencing experiments. Second, as both barcode and cDNA sequencing take a significant amount of the read length, how do the author envision they can be combined in future applications?

First, as reviewer #2 suggested, we replaced the length distribution in a histogram of 30 barcode sequencing (Figure 5 in this document) of the original Fig.1g. Second, we agree that the read length of our system is insufficient to obtain both barcode and cDNA sequence regions of individual cDNA molecules. In the future, we are designing an approach similar to Slide-seq, which was designed by adapting two sequencing platforms (SOLiD and Illumina). SOLiD was designed to decode barcode sequence and obtain spatial information, while both barcode and cDNA sequences of each molecule are analyzed with Illumina-sequencer. However, as we have described in the original manuscript, the spatial resolution achieved via SOLiD remains as subcellular resolution (not single-molecule level), thus we hope to improve the spatial resolution to single-molecule level with our system.

Figure 5. The length distribution in a histogram of 30 barcode sequencing

3. Also related to length: The authors state (page 6) that 73% of their reads aligned with the human genome. They also note that they considered every read longer than 4 nucleotides. What is the meaning of alignment of such short reads? A better approach would be to report how many reads they obtained longer than 18bp, and then how many of these aligned to the genome.

Thank you for your helpful suggestion. As the reviewer commented, we agree that reads shorter than 19 bp had little effect on detecting genes (number of detected genes). Thus, we re-analyzed the data using only reads longer than 18 bp, which resulted in a significant increase in the mapping rate (92.7% from 73%), while a slight decrease was observed in the number of detected genes (7148 from 8123). From this data we generated the results presented in the figures on the right, which replaced the original Fig.2.

Minor comments

1. Add a summary of the main sequencing errors with respect to the number of insertions, deletion and misincorporations. Are there any sequence dependent biases?

Thank you for the comment. In our original supplementary results (Fig. S2g), we had shown data aiming to summarize our sequencing errors, which indicated that the errors were primarily attributed to deletion errors. Besides this, we added the detailed analysis for sequencing errors, which we believe also provides insights to check sequence dependent biases (please see Fig. 4

New Figure 2 in the manuscript

in the response to reviewer #1). We found that, even when the same nucleic acid is incorporated, the accuracy of incorporation is slightly biased depending on the base immediately before the incorporation. We added these figures as the new figure S3.

2. Fig. 1A – The picture is not self-explanatory, denote the different parts of the system.

Thank you for this suggestion. We have added a more detailed description in Fig.1a. Also, we provided explanations for the images appearing in figure S12, please refer to the revised manuscript figures.

New Figure 1 in the manuscript

3. Fig. 1B – The illustration is ambiguous. It does not contribute to better understanding of the process. Add a more detailed explanation of the single molecule sequencing process or remove it.

Thank you for your suggestion. We modified the illustration presented in Fig.1b and provided a more detailed explanation in the corresponding figure legend as follows:

(B) Schematic illustration of single-molecule sequencing-by-synthesis utilized in our system. In each cycle, one of four virtual terminator nucleotides (VT-C, VT-T, VT-A, and VT-G) is incorporated (1), and unincorporated VTs are washed, followed by fluorescence imaging (2). The fluorescent dye and inhibitory groups are then removed using TCEP to permit the addition of the next virtual terminator (3). This cycle is repeated in the VT-C, VT-T, VT-A, VT-G order (4).

4. The synthesis of virtual terminators should be mentioned in the Methods section.

We have added the following text to the Methods section of the manuscript (Fill-and-lock step):

“We used virtual terminators supplied by Helicos or synthesized by Shinsei Chemical Company, Ltd. (Osaka, Japan). We outsourced synthesizing VTs to the chemical company according to the procedure described in the patent.²⁵”

5. What were the considerations in the barcode design?

We considered GC% of barcodes in the range of 13% to 62% to test detection biases. We did not observe a significant bias based on the GC%.

6. What were the BLAST parameters used for alignment?

Thank you for this point of clarification. We have provided additional details regarding barcode identification in the revised Methods section, including BLAST parameters, as follows:

Barcode identification

We mapped the sequence reads of 30 types of barcode molecules to the barcode ID reference (Table S1) using BLAST (2.2.30+) with the following option (blastn -db [barcode.reference] -query

[barode_read.fasta] -out [out.file] -word_size 4 -num_descriptions 1 -num_alignments 1 -dust no -strand plus -outfmt "7 std qlen qseq sseq"), and identified barcode IDs with BLAST E-values < 0.01.

7. Lines 182-184 – Elaborate more on the difference between the enzymes in the main text.

We have moved the description on the comparison of the enzymes originally described in the supplementary materials to the main text.

Reviewer #3 (Remarks to the Author):

In this work, Oguchi et al. propose restructuring existing sequencing method (HeliScope) to instead perform spatial resolution of DNA barcodes at single-molecule level. To demonstrate their approach, they first read 20 nt DNA sequences to spatially resolve 30 different DNA barcodes. They also demonstrate their approach in gene quantification context by trying to spatially resolve cDNA library for K562 cell. They also propose replacing prior flow cell binding chemistry with biotin-avidin which is a standard in microscopy. Finally, as a real-world application, they try to spatially identify DNA barcodes bound to antibodies to demonstrate that their method can be applied at the protein-level.

Usually there is a space v/s counts tradeoff as the authors correctly identify. If one desires to identify more types of barcodes, they can't be too close spatially and vice versa. The authors argue that since industry tool such as sequencers are proprietary, not much information is available on how they work. Therefore, to bridge the gap, they break apart Heliscopes, which is used routinely for DNA/RNA sequencing at scale, and instead re-purpose it to do single-molecule barcoding.

I like the overall idea in this work! Currently there are fancy barcoding techniques at single-molecule level but they all have their challenges. For example, geometric barcoding requires extensive preparation, temporal barcoding requires only one dye but longer data collection times, washing based techniques introduce additional drift in the sample, etc. Instead of trying to come up with a completely new technique to do single-molecule barcoding, why not simply use existing old sequencing machine lying in the lab since often times it is very difficult to setup a new barcoding technique in lab. Obviously then the question is how good will their method be? In this case, the authors show that it is reasonably versatile since they can do up to 30 barcodes with high accuracy and speed however they were all short strands (< 20 nt).

We appreciate your positive response toward our study concept.

Here are my comments:

1. Based on figure 1D, it seems like a lot of overlap between the individual barcodes in the fluorescence image. If there were two strands present in the same spot (or nearby), without super resolution imaging, it is impossible to determine whether the fluorescent blink came from which strand. Other studies such as DNA origami barcodes or temporal DNA barcodes have used concentration as low as 20 pM to avoid spatial overlap since biotin-avidin locations are non-specific on glass surface. The experiments sections say authors operated at 10 uM. The authors need to perform their sequencing experiments at a lot lower concentration. Even then they can't guarantee that more than one strands one sit in the same diffraction limited spot but the likelihood is a lot lower. Refer to these studies for the experimental protocol to minimize the spatial overlap of fluorescent spots and cite them appropriately.

a. Lin, Chenxiang, et al. "Submicrometre geometrically encoded fluorescent barcodes self-assembled from DNA." *Nature chemistry* 4.10 (2012): 832-839.

b. Shah, Shalin, Abhishek K. Dubey, and John Reif. "Programming temporal DNA barcodes for single-molecule fingerprinting." *Nano letters* 19.4 (2019): 2668-2673.

First of all, we note that, in order to show our sequencing performance at a single-molecule level, we had operated our sequencing run at a 25 pM, which is comparable to the concentration applied in the

references that the reviewer mentioned above. However, as the reviewer suggested, signal density might affect sequencing accuracy, thus we further analyzed the relationship between signal density and sequencing accuracy. First, we checked the signal density of data already shown in the original manuscript, which is ~ 0.4 molecules/ μm^2 . Regarding the signal density, by reducing the density of sample capture probes immobilized on a sequence flow cell, we also decreased the signal density. In fact, we obtained different signal densities ranging from ~ 0.1 to ~ 0.4 molecules/ μm^2 in six different conditions of capture probe density. Also, we estimated that ~ 0.1 molecules/ μm^2 (corresponding to ~ 3 μm in distance between molecules on average) is sufficiently sparse to detect signals as single-molecules even under diffraction-limited microscopy. Therefore, with these datasets, we performed a correlation analysis between the signal density and the error rate (Fig. 6 in this document). Here we calculated the error rate of the 2nd incorporation of two subsequent incorporations (Fig. 4 in this manuscript). Of 16 pairs of two subsequent incorporations, three pairs showed no significant correlation (Fig. 6d colored in grey), while the others exhibited weak correlations ($0.21 \leq r \leq 0.54$) with p -values < 0.05 . Thus, we conclude that the results presented in the original manuscript are less impacted by the signal density issue.

Figure 6. Relationship between the sequencing error rate and the signal density. Three examples of sequencing error rates of the 2nd incorporation in two subsequent incorporations against its signal density showing the highest correlation ($r = 0.54$), lowest correlation ($r = 0.21$) and no correlation in a, b and c, respectively. d) Summary of the correlation coefficient between the sequencing error rate and the signal density. Numbers in the heatmap indicate correlation coefficients, while black and grey values indicate statistical significance ($p < 0.05$) and insignificance ($p \geq 0.05$), respectively.

2. How exactly do authors do drift-correction is not very clear. They authors mention that they use Hatamatsu software but what algorithm do they use? Redundant cross-correlation, cross-correlation, Direct cross-correlation? One standard trick in imaging is to incubate sample on microscope stage for a few minutes before imaging starts to minimize thermal drift. I see that the authors write 2 – 4 minutes of incubation. It doesn't seem enough to me. Prior works have suggested 10 – 20 minutes of incubation time. Some imaging protocols:

- a. Roy, Rahul, Sungchul Hohng, and Taekjip Ha. "A practical guide to single-molecule FRET." *Nature methods* 5.6 (2008): 507-516.
- b. Schnitzbauer, Joerg, et al. "Super-resolution microscopy with DNA-PAINT." *Nature protocols* 12.6 (2017): 1198.

First, we would like to note that we used the software purchased from Hamamatsu Photonics for identifying molecule positions in each FOV ((x, y) coordination of each molecule in each FOV), not for directly conducting drift-compensations. Although the company did not disclose the details of the algorithm, they published a paper relating to this (Ref. 1, below). Regarding the drift-correction, we conducted this based on the molecule positions of fluorescence probes extracted via the software, not applying direct image correction (cross-correlation, etc.). We expected that molecule positions detected by each FOV should be overlapped to the corresponding position obtained at the 1st cycle, where we attempt to detect all molecules. We calculated transformation values of individual FOVs giving the maximum matching number to the 1st positions. To avoid confusion, we rewrote this section as follows:

Base calling (image analysis, stage drift compensation)

First, the positions of individual VT incorporations in each FOV were identified as pixel coordinates in integer values with software customized by Hamamatsu photonics.⁴² Next, we performed two rounds of the stage drift correction process. Note, we conducted this based on the pixel coordinates of individual VT incorporations identified via the software, not by applying a direct image correction process (such as cross-correlation). We expected that individual VT incorporations at each cycle would be overlapped to the corresponding position at the 1st cycle, where all molecules attempt to incorporate VTs. Thus, in the first round, the correction (translation) value of individual FOVs was determined so that the translated FOV shows maximum matching of molecules to those corresponding in the 1st cycle. Next, as the position markers, we extracted molecules at the 1st cycle showing the top 10% frequently observed (matched to) VT incorporations. In the second round, the correction (translation) value of individual FOVs was again determined so that the translated FOV indicated the maximum matching of molecules to "the position markers". Following drift compensation, individual VT incorporations matched to initial positions were identified with a tolerance of one pixel and transformed into sequence information (base calling). Reads that failed to cleave were excluded by examining the images after cleavage.

Ref.1. Takeshima, T., Takahashi, T., Yamashita, J., Okada, Y. & Watanabe, S. A multi-emitter fitting algorithm for potential live cell super-resolution imaging over a wide range of molecular densities. *J. Microsc.* 271, 266–281 (2018).

Regarding the incubation time before image acquisition, we did not require it to minimize the stage drifts. We added typical examples of stage drift for the same FOV below, showing that stage drift between adjacent intervals did occur but was not large (a few pixels, 145 nm/pix) and the whole drifts in a run remained within ~20 pixels.

Figure 7. Typical examples of stage drifts for the same FOV in imaging cycles. Each point indicates the stage drift value at each cycle from cycle #0 (1st position). These data were analyzed from 24Q individual runs including images of the cleaving process (4×24 images for VT incorporations and 4×24 images for cleaving). Approximate time scale is also indicated.

3. The authors in this work are trying to do single-molecule imaging with DNA to barcode (or uniquely identify them). However, I see no mention to the recent temporal DNA barcodes work or geometric barcodes work or washing based techniques etc. They should cite all the related references and add a small comparison (may be a table or paragraph) since the current work and prior works have same end-goal but are used in different contexts (fluorescence microscopy vs sequencing-by-imaging). For example, temporal barcode benefit is it uses only one dye, geometric barcode benefit is it can do 100s of barcodes, washing technique is useful because it can program 4^N barcodes, etc. While I see that the authors briefly talked about FISHseq+ in discussion that's only scratching surface. There a full field of DNA-based barcoding out there that is missed in the references section (mostly sequencing references). The authors should also look at these other works and cite them as they will find

that all these works have the same eventual goal (single-molecule barcoding).

- a. Lin, Chenxiang, et al. "Submicrometre geometrically encoded fluorescent barcodes self-assembled from DNA." *Nature chemistry* 4.10 (2012): 832-839.
- b. Johnson-Buck, Alexander, et al. "Kinetic fingerprinting to identify and count single nucleic acids." *Nature biotechnology* 33.7 (2015): 730-732.
- c. Jungmann, Ralf, et al. "Multiplexed 3D cellular super-resolution imaging with DNA-PAINT and Exchange-PAINT." *Nature methods* 11.3 (2014): 313-318.
- d. Shah, Shalin, Abhishek K. Dubey, and John Reif. "Programming temporal DNA barcodes for single-molecule fingerprinting." *Nano letters* 19.4 (2019): 2668-2673.
- e. Shah, Shalin, Abhishek K. Dubey, and John Reif. "Improved optical multiplexing with temporal DNA barcodes." *ACS Synthetic Biology* 8.5 (2019): 1100-1111.

We appreciate your suggestion. Accordingly, we have added the following paragraph to the beginning of the discussion.

“Fluorescence microscopy is a powerful tool for biological research, however, the ability to observe multiple objects simultaneously (multiplex) is limited by the number of spectrally distinguishable fluorophores. To overcome this limitation, several approaches have been devised to rely on DNA barcoding technologies, some of which offer simultaneous labeling of target molecules with orthogonal DNA barcoded affinity reagents¹, followed by sequential imaging via hybridization of dye-labeled complementary oligos.^{2,3} As an alternate approach temporal barcodes have been designed that do not rely on spectral information of the dye molecules but rather exploit distinct temporal fluorescence intensity signals produced via hybridization kinetics of dye-labeled complementary

oligos.^{4,6} Although this approach has significantly improved the multiplexing ability compared to conventional fluorescence microscopy, target specific probes are still required, which will ultimately limit the multiplex capacity of the system.

Furthermore, DNA barcoding technologies have also recently been applied for spatial transcriptome and proteome analyses. For this purpose, it is useful to decode barcode molecules using a sequencing-by-synthesis approach. For instance, (from here the original text continues) although CODEX is a fluorescence imaging-based technique, using DNA barcode molecule tagged antibodies, in place of conventional fluorophore tagged antibodies, and decoding them via a sequencing-by-synthesis offers highly multiplexed surface markers with which cells may be identified. Presently, these spatial resolutions correspond to specific cell sizes. However, our smSBS would allow visualization of individual molecules and increase the number of targets to be multiplexed (~~200~~) with a high degree of accuracy in identification, thereby advancing such analyses.”

1. Lin, C. et al. Submicrometre geometrically encoded fluorescent barcodes self-assembled from DNA. *Nat. Chem.* **4**, 832–839 (2012).
2. Jungmann, R. et al. Multiplexed 3D cellular super-resolution imaging with DNA-PAINT and Exchange-PAINT. *Nat. Methods* **11**, 313–318 (2014).
3. Woehrstein, J. B. et al. Sub-100-nm metafluorophores with digitally tunable optical properties self-assembled from DNA. *Sci. Adv.* **3**, 18–26 (2017).
4. Shah, S., Dubey, A. K. & Reif, J. Programming Temporal DNA Barcodes for Single-Molecule Fingerprinting. *Nano Lett.* **19**, 2668–2673 (2019).
5. Johnson-Buck, A. et al. Kinetic fingerprinting to identify and count single nucleic acids. *Nat. Biotechnol.* **33**, 730–732 (2015).
6. Shah, S., Dubey, A. K. & Reif, J. Improved Optical Multiplexing with Temporal DNA Barcodes. *ACS Synth. Biol.* **8**, 1100–1111 (2019).

4. Similar to drift-correction, I see that the flow chamber used by authors have 1x PBS buffer, other redox systems, and salts in their wash buffer. Usually it is not enough to avoid non-specific binding that occurs due to dye sticking on the surface and giving a false-positive. Did they do some treatment of glass surface before setting up flow chamber? Usually people use PEG for avoiding non-specific binding.

Thank you for the comment. As you pointed out, to avoid non-specific binding we used PEG-coated glasses as described in the Methods section.

5. Have the authors tried different capture times? I see that they use 200 ms exposure times, how did they come up with this since they will lose events faster than 200 ms and supposed if a base gets synthesized in 50 ms then they won't capture it in their read. Some explanation regarding this is required.

Thank you for the comment. First, we would like to note that the incorporation of nucleotides were NOT observed in “real-time”. The nucleotide incorporation and imaging steps were performed separately. In each nucleotide incorporation step, one of four types of fluorescently labeled nucleotides (virtual terminator) were introduced into a flow cell. In principle, only one base in a proper template should become extended. Then, after washing non-incorporated VTs, images were taken by each cycle. Regarding the exposure time, when we consider detection error (miss detection), a longer exposure would be preferable since the incorporated fluorescence probes are expected to be placed at the same position. Alternatively, a short exposure would be possible, however, it might require an increase in the excitation intensity, which would increase the risk of photobleaching rather resulting in detection errors. Of course, there is some trade-off between exposure time and excitation intensity for single molecule detection. Herein, we selected a 200 ms exposure time with an excitation intensity of 10 mW output at the laser head.

6. The authors on page 4 line 102 say that per quad they needed about 2 hours of data collection time and in total 2 days. This seems pretty slow. I was hoping that since they have multi-color dye and existing sequencer this should happen a lot faster. Can they talk about how they decided on the amount of data collection time?

To be honest, we prioritized mimicking the original system (Heliscope), while shortening the run time was not considered in the first trial. Although we believe that the reviewer grasped the details of our sequence-by-synthesis, we note that virtual terminators we used were originally invented by Helicos biosciences, and all four types of VTs are labeled with the same fluorescent dye (ATTO 647). Moreover, we outsourced synthesizing of VTs to the chemical company (Shinsei-kagaku, Japan) according to the procedure described in the patent. Of course, we might have been able to change the fluorescence dye instead of ATTO 647; however, according to the reference, they synthesized many candidates and carefully selected the one that allowed them to achieve accurate sequencing. Even a slight modification in the dye would affect overall properties of VTs, therefore, we decided to prioritize the reproducibility of the system conducting with a single fluorophore (besides this issue, it costs \$25,000 per VT synthesis, that is, \$100,000 in total). In the future, as the reviewer commented, it will be indispensable to shorten the analysis time, however, we expect that applying multi-color to the system will not be simple due to such circumstances. Hence, although our system required amendment to reduce the time requirements, we believe that our system offers a unique barcode decoding approach to the research community.

7. On page 5 line 112 the authors argue the readable length of a barcode increases with the increase in the number of sequencing cycles. While it sounds intuitively accurate, the figure 1f plot shows something else. It seems as if based on the average plot it nearly saturates after 20 nt i.e. at this point even if the data collection time is increased, one cannot read longer DNA. Is that correct? Can the authors comment on this? If so, they should talk about it deliberately in their discussion section as this is an important trade off. Since the total number of barcodes are 4^N where N is the length of barcode, learning how to accurately identify longer barcodes will be useful.

We partially agree with the reviewer's point. As shown in Fig. S2, some molecules stop the elongation reaction before reaching the final cycle (to date, we conducted up to a 24Q of sequence cycle). Besides this, we found that relatively substantial molecules (~43% of identified molecules) tolerated elongation at the final cycle, thus, we are not able to exclude the possibility that the molecules are lengthened in proportion to the number of cycles even after 24Q. However, we should also avoid stating that we expect we would be able to obtain longer read length as long as we repeat sequence cycles, and have, therefore, revised it as follows.

Original:

Efficiency of barcode molecule identification increased with the number of sequence cycles (Figure 1H), which, in turn, is attributed to the increase in read lengths in proportion to the number of sequence cycles.

Rephrased:

Efficiency of barcode molecule identification increased with the number of sequence cycles (Fig. 1h). ~~which, in turn, is attributed to the increase in read lengths in proportion to the number of sequence eyeles.~~

8. On page 9 line 222, the authors argue that overall by incorporating their technique the number of barcodes will scale up to 200. Can they please provide a reasoning to this? Some equation or calculation or justification on how they came up with this number since they show upto 30 in the paper and for figure 4g the efficiency is just slightly above 50%.

As described in the main text, in Fig.4, we examined our decoding ability with commercially available barcode molecules, with a 15 bp barcode sequence and ~200 types in the TotalSeqA lineup. We attempted to describe that can scale up to ~200 if we can extend the barcode length in such a way

as that shown in Fig. 1, instead of 15 bp. Also, as described in the response to reviewer #1, we were able to estimate the multiplexing ability with a simple model, indicating that our system would apply to decode a more barcode molecules (even for 10,000 unique molecules) than what we were able to show experimentally in this study. We have added this estimation to the revised Discussion section.

REVIEWERS' COMMENTS:

Reviewer #1 (Remarks to the Author):

I would like to thank the authors for carefully responding to all reviewer's comments. I have no other comments and would like to recommend this work for publication.

Reviewer #2 (Remarks to the Author):

The authors have adequately addressed my major concerns, and I therefore recommend this paper for publication.

Reviewer #3 (Remarks to the Author):

In this work, Oguchi et al. using sequencing technology to apply barcoding to spatial transcriptomics. In particular, they modify HeliScope tech to spatially decode DNA barcodes using their sequence at potentially a scale much smaller than diffraction limit of light (~ 250 nm). As a benchmark of their system, they sequenced with minimal error ~ 30 types of DNA strands (~ 20 nt long each). As an application of their system, they identified spatially DNA bound to antibodies on a cell surface hence supporting their claim that their method offers benefits to transcriptomics and proteomics.

Based on other reviews, it is pretty clear this manuscript is of interest to the broad audience of the journal. The manuscript's revision is also much more unified and cohesive. It connects well with the field of DNA nanoscience/computing, molecular biology/sequencing and single-molecular imaging which lacked in their v1. Based on the references suggested in the prior review (Shah et al. and Buck et al. and Lin et al.), I also see that the authors have added the "molecular footprinting" application which substantially increases the impact of their original manuscript. This idea is similar to carbon-copy or thumb impression using ink. However, by using a cell to show this instead of simply oligo on glass surface, it has a real single-molecule imaging application.

I see that a reviewer rightly pointed out that the system presented in this work is quite complex for labs without HeliScope tech and therefore it might not be as impactful. I do however, think that even though it is a complex system, some of the experimental tricks, protocols used, software tools, and cohesive introduction/references still makes the manuscript very interesting in other fields like DNA STORAGE where knowing sequencing limits is helpful. This study is more like "Towards single molecular FRET" protocol in that it tries several binding chemistry, calibrates sequencer, surface density of probes, etc.

Some minor edits mostly (again) to improve the readability, make clarification about what this method can and can't do, and cover references from wide variety of the field. Then I'd give a heads up for acceptance of this manuscript:

- There is one more reference that I'd like the authors to add: DNA microscopy since now with the additional application, they do talk about single-molecular imaging using sequencing. DNA microscopy is cool technique to reconstruct global cellular image from the local spatial information obtained using DNA sequencing. [https://www.cell.com/cell/pdf/S0092-8674\(19\)30547-1.pdf](https://www.cell.com/cell/pdf/S0092-8674(19)30547-1.pdf)

- In S11, "First, 10,000 barcode molecules were randomly sampled from the 30 barcode molecules with replacement, followed by error insertion to mimic the experimentally obtained sequence reads" => "First, we assumed a uniform distribution of 30 DNA barcodes ($\sim 4^{30} = 1e18$ possibilities) and sampled it, with replacement, 10,000 times to obtain i.i.d samples. Then we assumed that the sequencers reads each base as Bernoulli trials to mimic the error prone

experimental process of sequencing. This makes overall distribution of a DNA sample sequence Bernoulli where we assume success probability for each base p and number trials N (in this case 30 nt).” Then you can still use the full Bernoulli equation as further shows.

- “... which indicated that our system would be potentially applicable for up to 10,000 unique molecules” => since we have a very large number of samples mostly your system has higher capacity than 10k but I think major limitation currently is our existing tech hence you don't know if 10k is upper limit. It might be more. You can say something like several hundred or thousands. An example with reasonable errors for 10k sample that matches our empirical data distribution is shown in S11.

- Finally, the error at nucleotide is about 5% means with exponential decay in quality their error will soon grow to 90 – 100% which is okay. Just mention this clearly in discussion/ supplementary with a graph (error vs length of barcode). Because I think best synthesizers have under 1% error per base and they almost can't produce desired sequence after 100 – 120 nt. Also similar graph comparing it with sequencing cycle count. There is one is 1g and 2b showing cycle count and average read length as well as read length for 24Q. I want to see the limit under standard conditions. For example, if I want to implement your technique what max barcode length (to have max separation) should I use to make sure I still get reliable sequence reads without too many sequencing cycles? How should I design my barcodes? (Hint: free energy landscape of DNA hybridization might help)

Response to the Reviewer:

Reviewer #3 (Remarks to the Author):

In this work, Oguchi et al. using sequencing technology to apply barcoding to spatial transcriptomics. In particular, they modify HeliScope tech to spatially decode DNA barcodes using their sequence at potentially a scale much smaller than diffraction limit of light (~ 250 nm). As a benchmark of their system, they sequenced with minimal error ~ 30 types of DNA strands (~20 nt long each). As an application of their system, they identified spatially DNA bound to antibodies on a cell surface hence supporting their claim that their method offers benefits to transcriptomics and proteomics.

Based on other reviews, it is pretty clear this manuscript is of interest to the broad audience of the journal. The manuscript's revision is also much more unified and cohesive. It connects well with the field of DNA nanoscience/computing, molecular biology/sequencing and single-molecular imaging which lacked in their v1. Based on the references suggested in the prior review (Shah et al. and Buck et al. and Lin et al.), I also see that the authors have added the "molecular footprinting" application which substantially increases the impact of their original manuscript. This idea is similar to carbon-copy or thumb impression using ink. However, by using a cell to show this instead of simply oligo on glass surface, it has a real single-molecule imaging application.

I see that a reviewer rightly pointed out that the system presented in this work is quite complex for labs without HeliScope tech and therefore it might not be as impactful. I do however, think that even though it is a complex system, some of the experimental tricks, protocols used, software tools, and cohesive introduction/references still makes the manuscript very interesting in other fields like DNA STORAGE where knowing sequencing limits is helpful. This study is more like "Towards single molecular FRET" protocol in that it tries several binding chemistry, calibrates sequencer, surface density of probes, etc.

Some minor edits mostly (again) to improve the readability, make clarification about what this method can and can't do, and cover references from wide variety of the field. Then I'd give a heads up for acceptance of this manuscript:

Response: Thank you once again for your thorough evaluation of our manuscript. We appreciate all of your additional comments.

- There is one more reference that I'd like the authors to add: DNA microscopy since now with the additional application, they do talk about single-molecular imaging using sequencing. DNA microscopy is cool technique to reconstruct global cellular image from the local spatial information obtained using DNA sequencing. [https://www.cell.com/cell/pdf/S0092-8674\(19\)30547-1.pdf](https://www.cell.com/cell/pdf/S0092-8674(19)30547-1.pdf)

Response: We appreciate your suggestion. Accordingly, we have included the reference in our revised manuscript as follows:

"To overcome this limitation, several approaches have been devised by leveraging DNA barcoding technologies³¹⁻³⁷..."

31. Weinstein, J. A., Regev, A. & Zhang, F. DNA Microscopy: Optics-free Spatio-genetic Imaging by a Stand-Alone Chemical Reaction. *Cell* 178, 229-241.e16 (2019).

- In S11, "First, 10,000 barcode molecules were randomly sampled from the 30 barcode molecules with replacement, followed by error insertion to mimic the experimentally obtained sequence reads" => "First, we assumed a uniform distribution of 30 DNA barcodes (~ $4^{30} = 1e18$ possibilities) and sampled it, with replacement, 10,000 times to obtain i.i.d samples. Then we assumed that the

sequencers reads each base as Bernoulli trials to mimic the error prone experimental process of sequencing. This makes overall distribution of a DNA sample sequence Bernoulli where we assume success probability for each base p and number trials N (in this case 30 nt).” Then you can still use the full Bernoulli equation as further shows.

Response: Thank you for the comment. We would like to inform that Supplementary Fig. 11 conveys two different points. First, we validated the model analysis with the experimental results of 30 types of barcode molecules. For this validation, we randomly sampled 10,000 molecules from a unique distribution of 30 types of barcode molecules, followed by inserting sequencing errors in them and estimating the identification efficiency (New Supplementary Fig. 11c-f). After this validation, we considered a different case with 10,000 types of unique molecules (New Supplementary Fig. 11g and h). Thus, according to your advice, we have re-written the legend of Supplementary Fig. 11 as follows:

“... (c-f) Validation of the model analysis with the experimental identification of 30 barcode molecules. c) Generation of mock sequence reads for this simulation. First, 10,000 barcode molecules were randomly sampled from the 30 barcode molecules with replacement. Next, we assumed that the sequencing reads (each base, as per the Bernoulli trials) would mimic the error-prone experimental process of sequencing (d). The fraction of reads containing n errors ($P(n)$) should be governed by a binomial distribution ($P(n) = {}_N C_n (1-p)^n p^{(N-n)}$, where p : sequencing accuracy, N : read length, n : number of errors per read). Based on the distribution, we created mock sequencing reads by inserting errors (here, we simply inserted substitution errors) into the original sequences. e) Identification of mock sequencing reads by aligning them to the original 30 barcode sequences with BLAST. f) Comparison of simulated identification efficiencies with the experimentally obtained result. The result of simulation with $p = 0.85$ provides a suitable fit for the experiment. (g, h) Estimation of the identification efficiency of 10,000 types of unique molecules. Briefly, we estimated an identification efficiency of 10,000 types of unique molecules with read lengths ranging from 15 to 30 bp, with $p = 0.8$ (g). First, we sampled 10,000 molecules from a uniform distribution of DNA barcodes 30 nt in length ($\sim 4^{30}$ possibilities) without replacement, followed by error insertion to create mock sequencing reads. The mock sequencing reads of the 10,000 types of unique molecules were identified via alignment with the original sequences using BLAST.”

- “... which indicated that our system would be potentially applicable for up to 10,000 unique molecules” => since we have a very large number of samples mostly your system has higher capacity than 10k but I think major limitation currently is our existing tech hence you don't know if 10k is upper limit. It might be more. You can say something like several hundred or thousands. An example with reasonable errors for 10k sample that matches our empirical data distribution is shown in S11.

Response: We appreciate your suggestion. We have re-written the sentence in the revised main text as follows:

“Although in the current study we demonstrated that our system effectively multiplexes up to 30 targets, we were able to further estimate the multiplexing capacity using a simple model (Supplementary Fig. 11a-f). This indicated that our system could be potentially applicable for several hundred or thousand unique molecules. An example (with reasonable errors) for 10,000 unique molecules matching our empirical data distribution is shown in Supplementary Fig. 11g and h.”

- Finally, the error at nucleotide is about 5% means with exponential decay in quality their error will soon grow to 90 – 100% which is okay. Just mention this clearly in discussion/ supplementary with a graph (error vs length of barcode). Because I think best synthesizers have under 1% error per base and they almost can't produce desired sequence after 100 – 120 nt. Also similar graph comparing it with sequencing cycle count. There is one is 1g and 2b showing cycle count and average read length as well as read length for 24Q. I want to see the limit under standard conditions. For example, if I

want to implement your technique what max barcode length (to have max separation) should I use to make sure I still get reliable sequence reads without too many sequencing cycles? How should I design my barcodes? (Hint: free energy landscape of DNA hybridization might help)

Response: We appreciate your comments. As per your suggestion, we have added a graph showing the fraction of the number of errors in a read against the read length as the new Supplementary Fig. 11a. Additionally, we have added the error distribution in a read (experimentally obtained) as the new Supplementary Fig. 11b. Moreover, we have updated the legend of Supplementary Fig. 11, accordingly, as follows:

“... (a, b) The number of errors in a sequence read was obtained experimentally. The fraction of error containing reads (having at least one error) increased with the increase in read length (a). Distribution of the number of errors in a read for 24Q (b).”

Furthermore, we appreciate your question. We were also interested in determining the maximum barcode length that would allow the maximum separation. As per the analysis shown in Supplementary Fig. 11g and h, the identification efficiency seems to increase with the increase in read length. However, to obtain a specific length, we need to perform further experiments in the future.

Supplementary Fig. 11: Model analysis estimating the identification efficiency of barcode molecules. (a, b) The number of errors in a sequence read was obtained experimentally. The fraction of error containing reads (having at least one error) increased with the increase in read length (a). Distribution of the number of errors in a read for 24Q (b). (c-f) Validation of the model analysis with the experimental identification of 30 barcode molecules. c) Generation of mock sequence reads for this simulation. First, 10,000 barcode molecules were randomly sampled from the 30 barcode molecules with replacement. Next, we assumed that the sequencing reads (each base, as per the Bernoulli trials) would mimic the error-prone experimental process of sequencing (d). The fraction of reads containing n errors ($P(n)$) should be governed by a binomial distribution ($P(n) = {}_N C_n (1-p)^n p^{(N-n)}$, where p : sequencing accuracy, N : read length, n : number of errors per read). Based on the distribution, we created mock sequencing reads by inserting errors (here, we simply inserted substitution errors) into the original sequences. e) Identification of mock sequencing reads by aligning them to the original 30 barcode sequences with BLAST. f) Comparison of simulated identification efficiencies with the experimentally obtained result. The result of simulation with $p = 0.85$ provides a suitable fit for the experiment. (g, h) Estimation of the identification efficiency of 10,000 types of unique molecules. Briefly, we estimated an identification efficiency of 10,000 types of unique molecules with read lengths ranging from 15 to 30 bp, with $p = 0.8$ (g). First, we sampled 10,000 molecules from a uniform distribution of DNA barcodes 30 nt in length ($\sim 4^{30}$ possibilities) without replacement, followed by error insertion to create mock sequencing reads. The mock sequencing reads of the 10,000 types of unique molecules were identified via alignment with the original sequences using BLAST. The analysis with an e-value < 0.01 shows almost no incorrect identifications; therefore, a looser threshold, such as an e-value < 0.1 , can be applied, resulting in an increased identification efficiency, especially for 15 bp read length sequences (h). The process of identification of molecules is described in e.